# Nedd4 ubiquitylates VDAC2/3 to suppress erastin-induced ferroptosis in melanoma

Yongfei Yang [1,2,4]*, Meiying Luo[1,2,4], Kexin Zhang [1,4], Jun Zhang[2], Tongtong Gao[1], Douglas O' Connell [3], Fengping Yao[1], Changwen Mu [1], Bingyu Cai [1], Yuxue Shang [1] & Wei Chen[2]*

Ferroptosis is a newly defined form of regulated cell death characterized by the iron-dependent accumulation of lipid hydroperoxides. Erastin, the ferroptosis activator, binds to voltage-dependent anion channels VDAC2 and VDCA3, but treatment with erastin can result in the degradation of the channels. Here, the authors show that Nedd4 is induced following erastin treatment, which leads to the ubiquitination and subsequent degradation of the channels. Depletion of Nedd4 limits the protein degradation of VDAC2/3, which increases the sensitivity of cancer cells to erastin. By understanding the molecular mechanism of erastin-induced cellular resistance, we can discover how cells adapt to new molecules to maintain homeostasis. Furthermore, erastin-induced resistance mediated by FOXM1-Nedd4-VDAC2/3 negative feedback loop provides an initial framework for creating avenues to overcome the drug resistance of ferroptosis activators.

---

[1] Key Laboratory of Molecular Medicine and Biotherapy, School of Life Science, Beijing Institute of Technology, Beijing 100081, China. [2] Beijing Institute of Biotechnology, Beijing 100071, China. [3] College of Osteopathic Medicine, Touro University, Vallejo, CA 94592, USA. [4]These authors contributed equally: Yongfei Yang, Meiying Luo, Kexin Zhang *email: yangyf@bit.edu.cn; cw0226@foxmail.com

C ell death is critical in diverse aspects of mammalian development, homeostasis, and disease. In recent years, several distinct forms of programmed cell death (PCD) have been identified with each having unique cellular mechanisms. Ferroptosis is a newly recognized form of PCD, which is readily distinguishable from other types of PCD such as apoptosis, necroptosis, and autophagic cell death at morphological, biochemical, and genetic levels[1,2]. Recent studies have demonstrated that not only the primary cells but also drug-resistant cells from different cancers are especially sensitive to ferroptosis[3–8]. Moreover, most notably in malignant melanoma, where both receptor tyrosine kinase-mediated resistance to mitogen-activated protein kinase targeted therapies and activation of the inflammatory signaling associated with immune therapy lead to increased sensitivity to ferroptosis[5]. These works overcome reactionary resistance by developing strategies to destroy cancer cells by inducing ferroptotic cell death.

The voltage-dependent anion channel (VDAC) is a pore located at the outer membrane of the mitochondrion. It allows the entry and exit of numerous ions and metabolites between the cytosol and the mitochondrion. By managing the exchange of molecules between cellular compartments, VDAC regulates various cellular processes like apoptosis, metabolism, ion homeostasis, and thus, impacts many diseases including cancer[9]. Erastin, the first identified ferroptosis activator in 2003, was found to exhibit substantial lethality in human tumor cells harboring mutations in oncogenes HRAS, KRAS, and BRAF[10]. It has been reported that erastin induces ferroptosis through directly binding to VDAC2/3 to alter the permeability of the outer mitochondrial membrane, which decreases the rate of NADH oxidation[10]. Besides exerting targeted effects, erastin also enhances chemotherapy, targeted therapy, and immunotherapy in certain cancer cells[5,11,12], suggesting a potential role of erastin in cancer cell treatment. However, after 10 h of erastin treatment, both VDAC2 and VDAC3 were no longer detectable, and this low expression level of VDAC2/3 led to erastin resistance[10]. The mechanism by which erastin induces specific downregulation of VDAC2/3 has not been fully elucidated.

Neuronal precursor cell-expressed developmentally downregulated 4 (Nedd4) is an essential member of HECT domain E3 ligase family in eukaryotes and remains conservative during the evolution. It is comprised of a catalytic C-terminal HECT domain, an N-terminal calcium/lipid-binding domain (C2 domain), and four WW domains responsible for cellular localization and substrate recognition[13]. Nedd4 plays a pivotal role in numerous biological processes through proteasomal degradation of its substrates that generally have a PPxY motif for WW domain recognition, and are found in the nucleus or at the plasma membrane[13]. Here, we report that erastin activates the transcriptional expression of Nedd4 through FOXM1 in melanoma cells. As an E3 ligase, Nedd4 binds to the PPxY motifs of VDAC2/3 through its WW domain and degrades them. Downregulation of Nedd4 by shRNA rescued erastin-induced protein elimination of VDAC2/3 and increased the sensitivity of melanoma cells to erastin. Together, we identify an E3 ligase that regulates ferroptosis in melanoma cells and provide further insight into Nedd4 as a target for overcoming erastin-induced resistance to cancer therapy.

## Results

**Erastin degrades VDAC2/3 in melanoma cells**. To test whether erastin can degrade VDAC2/3 in melanoma cells, we transfected A375 melanoma cells bearing the BRAF[V600E] mutation with GFP-VDAC2 or GFP-VDAC3 and mCherry-VDAC1. Upon erastin treatment, the fluorescence intensity levels of VDAC2 and VDAC3 were sharply reduced, but the level of VDAC1 was only slightly reduced (Fig. 1a). Immunoblotting of endogenous VDAC1, VDAC2, and VDAC3 with their specific antibodies (Supplementary Fig. 1a) showed similar results in erastin treated A375 and G-361 cells, both of which carry BRAF[V600E] mutation (Fig. 1b and Supplementary Fig. 1b). We further examined whether knockdown of VDAC2/3 suppresses erastin-induced ferroptosis in melanoma cells. As shown in Fig. 1c, d, depletion of VDAC2 or VDAC3 by RNAi resulted in significantly increased resistance to erastin compared to control RNAi. Indeed, suppression of VDAC2 or VDAC3 significantly inhibited erastin-induced ferroptotic events, including lipid ROS production, iron accumulation, glutathione (GSH) depletion, and glutathione disulfide (GSSG) generation (Fig. 1c, d). Moreover, combined interference with VDAC2 and VDAC3 showed a stronger effect on lipid ROS production and iron accumulation (Fig. 1c, d). Overexpression of VDAC2 or VDAC3 individually did not affect erastin-induced ferroptotic cell death, but VDAC2 and VDAC3 together facilitated erastin-induced ferroptosis in A375 cells (Fig. 1e, f). Together, these results indicate that both VDAC2 and VDAC3 are essential for erastin-induced ferroptosis in A375 cells, and these results are consistent with previous work in engineered human tumor cells (BJTERT/LT/ST/RAS[V12])[10].

We next detected whether the reduction of VDAC2/3 was regulated at the transcriptional level or posttranscriptional level. The mRNA level of VDAC2/3 was not reduced in erastin treated A375 and G-361 cells (Supplementary Fig. 1c), indicating that erastin regulates the expression of VDAC2/3 post transcriptionally. To test this hypothesis, we treated cells with either MG132, a proteasome inhibitor, or $NH_4Cl$, a lysosome inhibitor, and found that the protein level of VDAC2/3 was sensitive to MG132, but not to $NH_4Cl$ (Fig. 1g), suggesting that VDAC2/3 is regulated through the proteasome pathway. Consistent with this observation, erastin treatment increased the ubiquitination of Myc-VDAC2 and Myc-VDAC3 on the K48-linked ubiquitin chain, but not the K63-linked ubiquitin chain (Fig. 1h and Supplementary Fig. 1d). Collectively, our data illustrate that erastin triggers ubiquitin–proteasomal degradation of VDAC2/3 in melanoma cells.

**Nedd4 directly recognizes VDAC2/3**. We next investigated which E3 ubiquitin ligase mediated the protein degradation of VDAC2/3 induced by erastin. On examination of the protein sequences of VDAC2/3, we surmised a proline-rich PPxY motif that can be recognized by Nedd4 (Fig. 2a). Interestingly, The PPxY motifs of VDAC2/3 are highly conserved among different species (Supplementary Fig. 2a, b), although the sequence of VDAC3 evolved from PPxY to TPxY (Supplementary Fig. 2b). Consistently, Nedd4 is predicted with the highest confidence as a primary E3 ligase for VDAC2/3 in UbiBrowser database (Supplementary Fig. 2c, d).

We then confirmed the interaction between endogenous VDAC2/3 and Nedd4 by co-immunoprecipitation (IP) experiments, and a stronger interaction was detected after erastin treatment (Fig. 2b), suggesting that Nedd4 associates with VDAC2/3 in A375 cells. We further performed an in vitro pulldown assay using glutathione S-transferase (GST)-Nedd4 purified from *Escherichia coli* and immunopurified Flag-VDAC2 and Flag-VDAC3 proteins. Both VDAC2 and VDAC3 were readily detected in the fractions eluted from the GST-Nedd4 affinity column but not in elutes from the GST column, indicating that the interaction between these proteins was direct (Fig. 2c). Moreover, the PPxY/TPxY motif mutations of VDAC2 and VDAC3 abolished the interactions with Nedd4 (Fig. 2d), and the WW domain of Nedd4 was crucial for binding to VDAC2/3

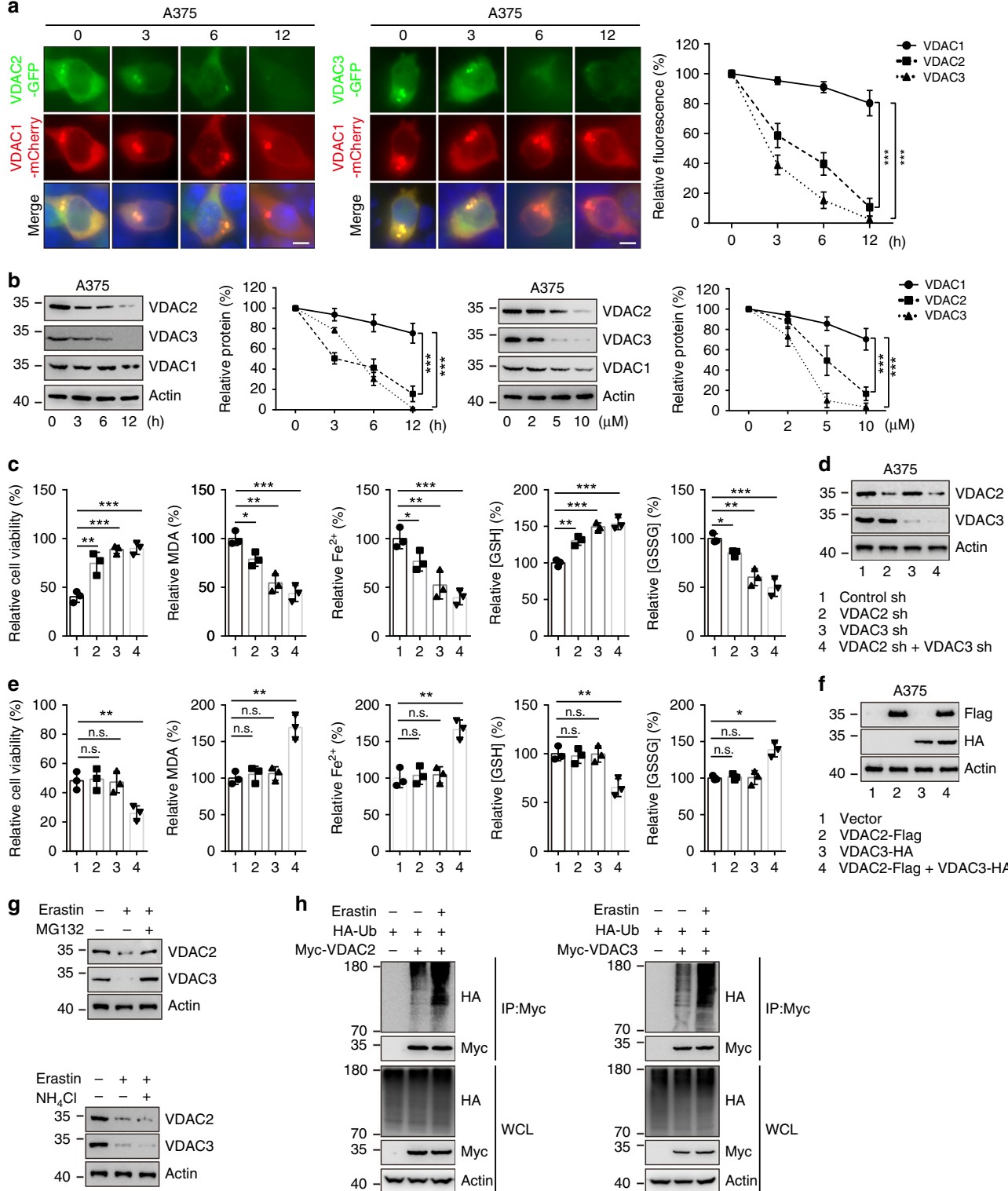

(Fig. 2e, f), which were similar to other identified substrates. Taken together, our data suggest that Nedd4 binds to the PPxY/ TPxY motif of VDAC2/3 through its WW domain.

**Nedd4 ubiquitinates and degrades VDAC2/3.** To test whether Nedd4 affects the cellular level of VDAC2/3, we overexpressed wild-type (wt) Nedd4 in A375 cells and found that the endogenous protein level of VDAC2/3 was sharply reduced (Fig. 3a). However, ectopic expression of Nedd4$^{C867S}$, which

lacks ubiquitin ligase activity, did not affect the level of VDAC2/3, indicating that the E3 catalytic activity of Nedd4 was required for VDAC2/3 protein destabilization (Fig. 3a). Consistently, the half-life of VDAC2/3 was significantly reduced in Nedd4 overexpression cells (Supplementary Fig. 3a) but not in Nedd4$^{C867S}$ overexpression cells (Supplementary Fig. 3b) as detected by cycloheximide chase assay. These results suggest that Nedd4 is the E3 ligase that destabilizes VDAC2/3 in melanoma cells.

**Fig. 1 Erastin treatment promotes ubiquitin–proteasomal degradation of VDAC2/3. a** A375 cells expressing GFP-VDAC2 or GFP-VDAC3 and mCherry-VDAC1 were treated with erastin (5 μM). Representative images were taken at the indicated time after treatment. Scale bars, 10 μm. Fluorescence intensity was assayed using Image J software. **b** A375 cells were treated with erastin at the indicated time (0–12 h, 10 μM) or indicated concentrations (0–10 μM, 12 h). Western blot analysis shows the expression of VDAC2, VDAC3, and VDAC1 in indicated cells. **c, d** Knockdown of VDAC2 and VDAC3 individually or in combination suppressed erastin-induced ferroptotic cell death. Indicated cells were treated with erastin (5 μM) for 24 h. Cell viability was assayed using a CCK8 kit. The lipid formation was measured by MDA assay. The accumulation of $Fe^{2+}$ was measured by iron detection assay. The concentrations of GSH and GSSG were detected by relative assay kits (**c**). The protein level of VDAC2/3 was assessed by immunoblotting (**d**). **e, f** Overexpression of VDAC2 and VDAC3 together increased erastin-induced ferroptotic cell death. Indicated cells were treated with erastin (5 μM) for 24 h. Cell viability, intracellular MDA, $Fe^{2+}$, GSH, and GSSG levels were measured (**e**). The overexpression of VDAC2/3 was assessed by immunoblotting (**f**). **g** A375 cells were treated with erastin (5 μM) for 8 h, then MG132 (50 mM) or $NH_4Cl$ (50 mM) were added into the culture medium for 4 h. The protein level of VDAC2/3 was analyzed by western blot. **h** A375 cells were transfected with indicated DNA constructs. After 48 h transfection, cells were treated with DMSO or erastin (5 μM) for 8 h, then MG132 (50 mM) was added into the culture medium for 4 h. Lysates from A375 cells were immunoprecipitated with mouse anti-Myc. Western blots were performed to analyze the presence of indicated proteins and levels of ubiquitination. Actin was used as a loading control. Data shown represent mean ± SD from three independent experiments. Comparisons were made using Student's $t$ test. *$p < 0.05$; **$p < 0.01$; ***$p < 0.001$.

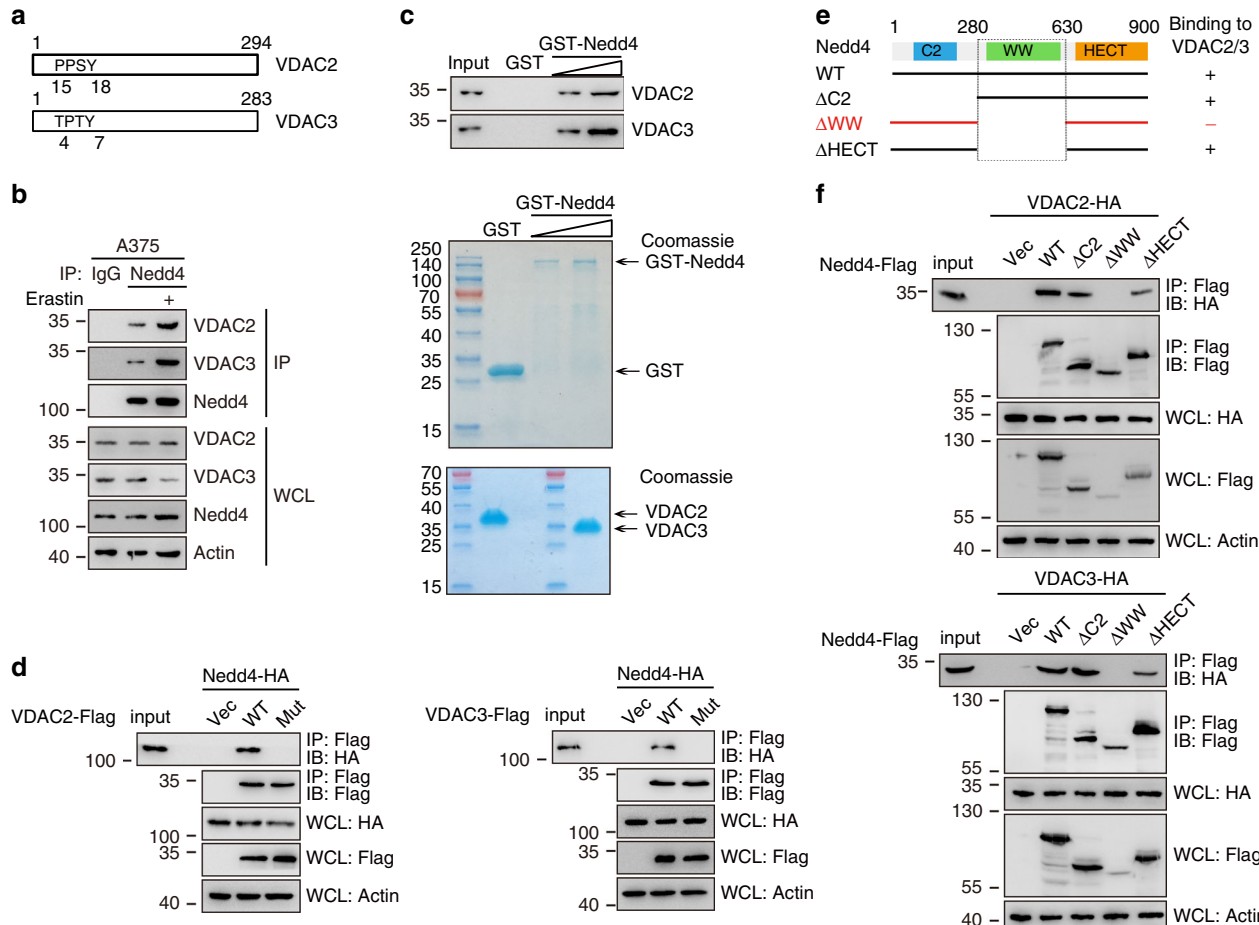

**Fig. 2 Identification of Nedd4 as a VDAC2/3 interacting protein. a** The PPxY motifs of VDAC2 (PPSY) and VDAC3 (TPxY) are conserved among different species. See also Fig. S2A, B. **b** Interaction between endogenous VDAC2, VDAC3, and Nedd4 under basal condition and erastin treatment. A375 cells were treated with DMSO or erastin (5 μM) for 4 h, then MG132 (50 mM) was added into the culture medium for an additional 4 h. Whole-cell lysates (WCL) were used for IP with control serum (IgG) or anti-Nedd4 antibody, followed by immunoblotting (IB) with the indicated antibodies. **c** Nedd4 directly interacts with VDAC2/3 in vitro. Purified GST (control) or increasing amount of GST-Nedd4 were mixed with VDAC2 or VDAC3 and subjected to GST-pulldown followed by IB for VDAC2 or VDAC3. Efficient VDAC2/VDAC3-Nedd4 interaction was detected in a dose-dependent manner. Coomassie blue stains show the input of each purified proteins used for binding. **d** The PPxY mutants of VDAC2 and VDAC3 are defective in Nedd4 binding. A375 cells were transfected with indicated DNA constructs for 48 h and treated with MG132 (50 mM) for 4 h. WCL were immunoprecipitated with anti-Flag followed by IB with anti-HA. **e** Schematic representation of WT Nedd4 and its deletion mutants, and summary of their interactions with VDAC2/3. Plus indicates strong binding; minus indicates no binding. **f** The WW domain of Nedd4 is essential for interaction with VDAC2/3. HA-VDAC2 or HA-VDAC3 was cotransfected with various Flag-Nedd4 deletion mutants in HEK293T cells. At 24 h post transfection, cells were treated with 50 mM MG132 for 4 h. WCL were subjected to immunoprecipitation with an anti-Flag antibody, followed by IB with anti-Flag or anti-HA. WCL were also used for IB with the indicated antibodies to show expression.

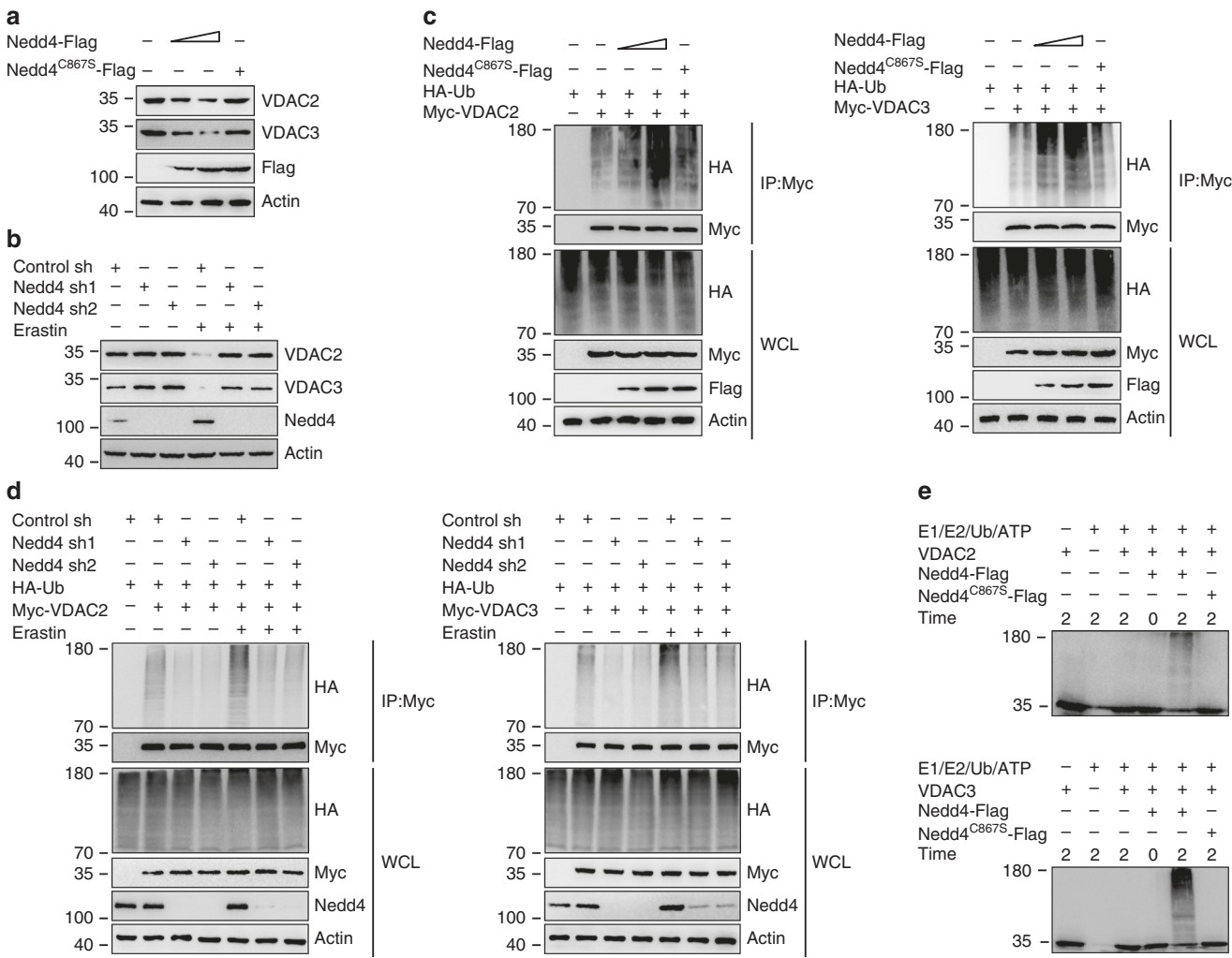

**Fig. 3 Nedd4 negatively regulates VDAC2/3 stability as the specific E3 ubiquitin ligase. a** Nedd4 decreased VDAC2/3 protein in a dose-dependent manner. A375 cells were transfected with Flag-Nedd4 (0, 1.5, and 6 μg) or Flag-Nedd4$^{C867S}$ (6 μg). The protein expression level of VDAC2/3 was assayed by western blot. Nedd4$^{WT}$ can destabilize VDAC2/3, but Nedd4$^{C867S}$ cannot affect the stability of VDAC2/3. **b** Knockdown of Nedd4 stabilizes VDAC2/ 3. A375 cells were transfected with control shRNA or Nedd4 shRNAs for 36 h, then treated with DMSO or Erastin (5 μM) for 12 h. The protein levels of VDAC2, VDAC3, and Nedd4 were analyzed by western blot. **c** Nedd4 ubiquitylates VDAC2/3 in vivo. A375 cells were transfected with indicated DNA constructs for 48 h and treated with MG132 (50 mM) for 4 h before harvest. Cell lysates were immunoprecipitated with anti-Myc and analyzed by immunoblotting with indicated antibodies. **d** Knockdown of Nedd4 reduced the ubiquitination of VDAC2/3 in vivo. A375 cells were transfected with indicated DNA constructs for 36 h, then treated with DMSO or erastin (5 μM) for 8 h. Before cell harvest, MG132 (50 mM) was added into the medium for 4 h. Cell lysates were immunoprecipitated with anti-Myc and analyzed by immunoblotting with indicated antibodies. **e** Nedd4 ubiquitylates VDAC2/3 in vitro. Purified VDAC2 and VDAC3 proteins were ubiquitylated in the presence of purified Nedd4 in vitro. See "Methods" for further details. After in vitro ubiquitylation reaction, samples were analyzed by immunoblotting with anti-VDAC2 and anti-VDAC3 antibodies.

To investigate whether endogenous Nedd4 contributes to the erastin-induced protein degradation of VDAC2/3, we transfected A375 cells with two shRNA directed against Nedd4. Depletion of Nedd4 resulted in a slight increase in the amount of VDAC2/3, and the effect of Nedd4 was more substantial after erastin treatment (Fig. 3b). Consistently, knockdown of Nedd4 extended the half-life of VDAC2/3, and the effect of Nedd4 was more significant after erastin treatment (Supplementary Fig. 3c). Next, we investigated whether Nedd4 promotes ubiquitination of VDAC2/3. As shown in the ubiquitination assays, overexpression of Nedd4 significantly increased the K48-linked ubiquitination of VDAC2/3, but Nedd4$^{C867S}$ did not (Fig. 3c and Supplementary Fig. 3d). Consistent with these observations, we found that knockdown of Nedd4 markedly reduced the ubiquitination of VDAC2/3 in A375 cells (Fig. 3d). Further, VDAC2/3 purified from *E. Coli* was ubiquitylated in vitro upon incubation with

bacteria-expressed Nedd4, but not Nedd4$^{C867S}$ (Fig. 3e). Taken together, these results demonstrate that Nedd4 directly binds to and ubiquitylates VDAC2/3.

**Nedd4 negatively regulates erastin-induced ferroptosis.** Given that Nedd4 binds to and degrades VDAC2/3 in erastin treated A375 cells, we next elucidated the function of Nedd4 in ferroptosis. Suppression of Nedd4 by specific shRNA promoted erastin-induced cell death in A375 and G361 cells (Fig. 4a), along with increased ferroptotic events including lipid ROS production, iron accumulation, GSH depletion, and GSSG generation (Fig. 4b–f). Erastin sensitivity was abolished by reexpression of shRNA-resistant Nedd4 (Fig. 4a–f). In contrast, reexpression of Nedd4$^{C867S}$ mutant failed to confer erastin resistance (Fig. 4a–f). Likewise, ectopic expression of Nedd4, but not Nedd4$^{C867S}$,

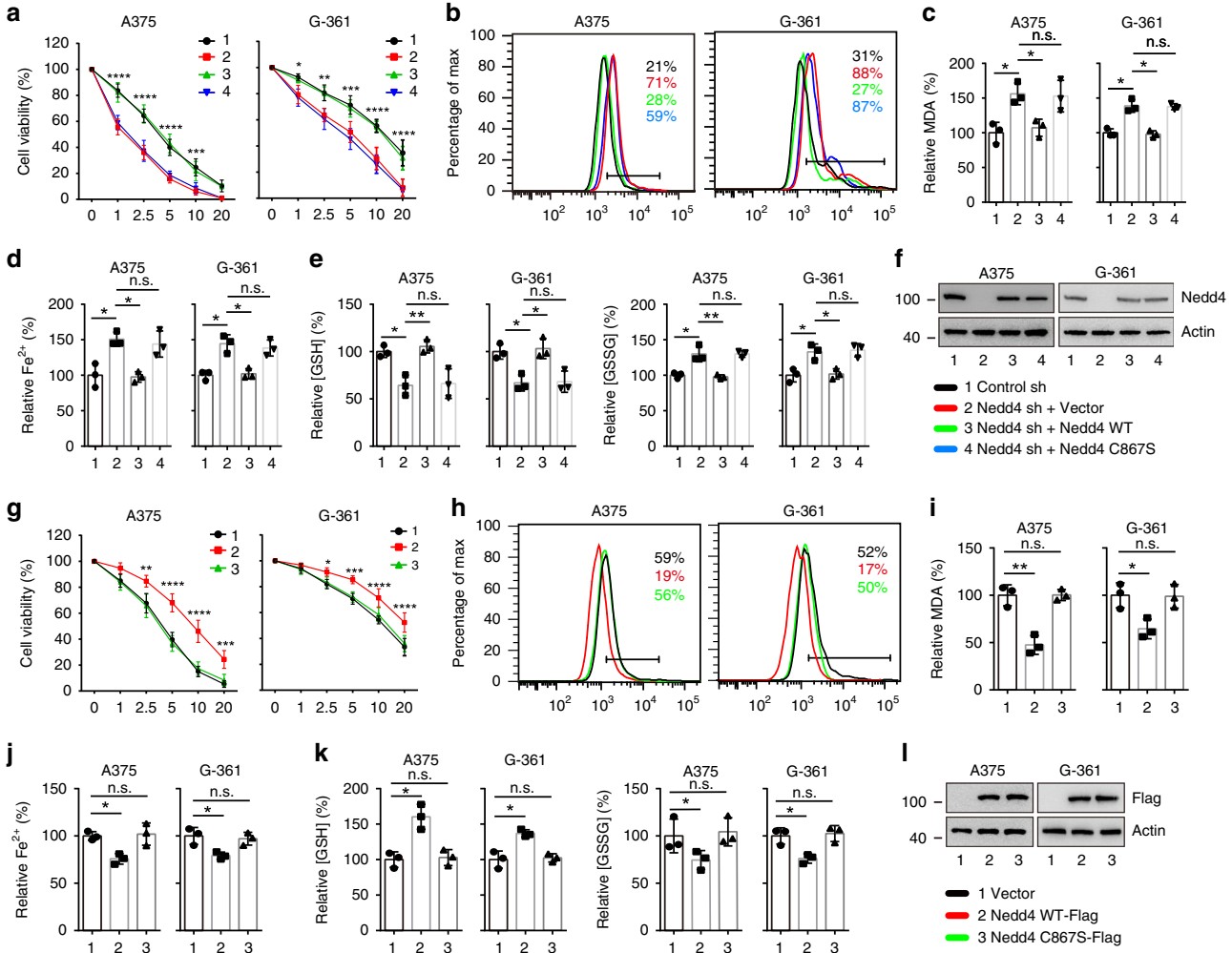

**Fig. 4 Nedd4 regulates erastin-induced ferroptosis. a–f** Knockdown of Nedd4 enhanced erastin-induced ferroptotic cell death. Indicated cells were treated with erastin (1–20 μM) for 24 h, and cell viability was assayed using a CCK8 kit (**a**). Indicated cells were treated with erastin (5 μM) for 24 h, the lipid ROS level was assessed by flow cytometry using C11-BODIPY (**b**), and the lipid formation was measured by MDA assay (**c**). The accumulation of $Fe^{2+}$ was measured by iron detection assay (**d**), concentrations of GSH, and GSSG were detected by relative assay kits (**e**). The expression of Nedd4 was assessed by immunoblotting and compared to actin levels (**f**). Number 1 and black color represent control sh group; number 2 and red color represent Nedd4 sh + Vector group; number 3 and green color represent Nedd4 sh + Nedd4 WT group; number 4 and blue color represent Nedd4 C867S group. **g–l** Overexpression of Nedd4 suppressed erastin-induced ferroptotic cell death. Indicated cells were treated with erastin (1–20 μM) for 24 h, and cell viability was assayed using a CCK8 kit (**g**). Indicated cells were treated with erastin (5 μM) for 24 h, the lipid ROS level was assessed by flow cytometry using C11-BODIPY (**h**), and the lipid formation was measured by MDA assay (**i**). The accumulation of $Fe^{2+}$ was measured by iron detection assay (**j**), concentrations of GSH and GSSG were detected by relative assay kits (**k**). The expression of Nedd4 was assessed by immunoblotting and compared to actin levels (**l**). Number 1 and black color represent vector group; number 2 and red color represent Nedd4 WT-Flag group; number 3 and green color represent Nedd4 C867S-Flag group. Data shown represent mean ± SD from three independent experiments. Comparisons were made using Student's $t$ test and one-way ANOVA. *$p < 0.05$; **$p < 0.01$; ***$p < 0.001$; ****$p < 0.0001$; n.s. not significant.

caused increased cell viability in erastin treated A375 and G-361 cells, due to reduced ferroptotic events (Fig. 4g–l). Ferrous iron ($Fe^{2+}$) and lipid ROS are essential for ferroptosis. The iron chelators, deferoxamine (DFO) and ciclopirox (CPX), can protect cells from erastin-induced ferroptosis in the presence of Nedd4 overexpression (Supplementary Fig. 4a–d). Similarly, the scavengers for lipid ROS, ferrostatin-1 (Fer-1), and Liproxstatin-1 (Lip-1), limit erastin-induced ferroptosis in Nedd4 overexpression cells (Supplementary Fig. 4e–h). RSL3 is another ferroptosis activator by targeting glutathione peroxidase 4 (GPX4) which is a critical antioxidant enzyme. Intriguingly, knockdown of FOXM1 or Nedd4 suppressed RSL3-induced protein degradation of VDAC2/3 (Supplementary Fig. 5a). However, neither downregulation nor overexpression of Nedd4 had a significant effect on RSL3-induced ferroptosis in

melanoma cells (Supplementary Fig. 5b–g). These results indicate that Nedd4 specifically regulates erastin-induced ferroptosis in melanoma cells.

**Nedd4 ubiquitinates VDAC2/3 in a site-specific manner.** All three isoforms of the VDAC family have similar protein sequences (83% similarity and 67% identity). Similar to VDAC2/3, VDAC1 also has a Nedd4 interacting PPxY motif, which is conserved among different species (Supplementary Fig. 6a). Nedd4 was predicted as a major E3 ligase for mediating the degradation of VDAC1 (Supplementary Fig. 6b). As expected, endogenous VDAC1 co-immunoprecipitated with Nedd4 in A375 cells, and this interaction was enhanced by erastin treatment (Supplementary Fig. 6c). However, as shown in Fig. 1, erastin mainly degrades VDAC2/3 but not VDAC1, suggesting

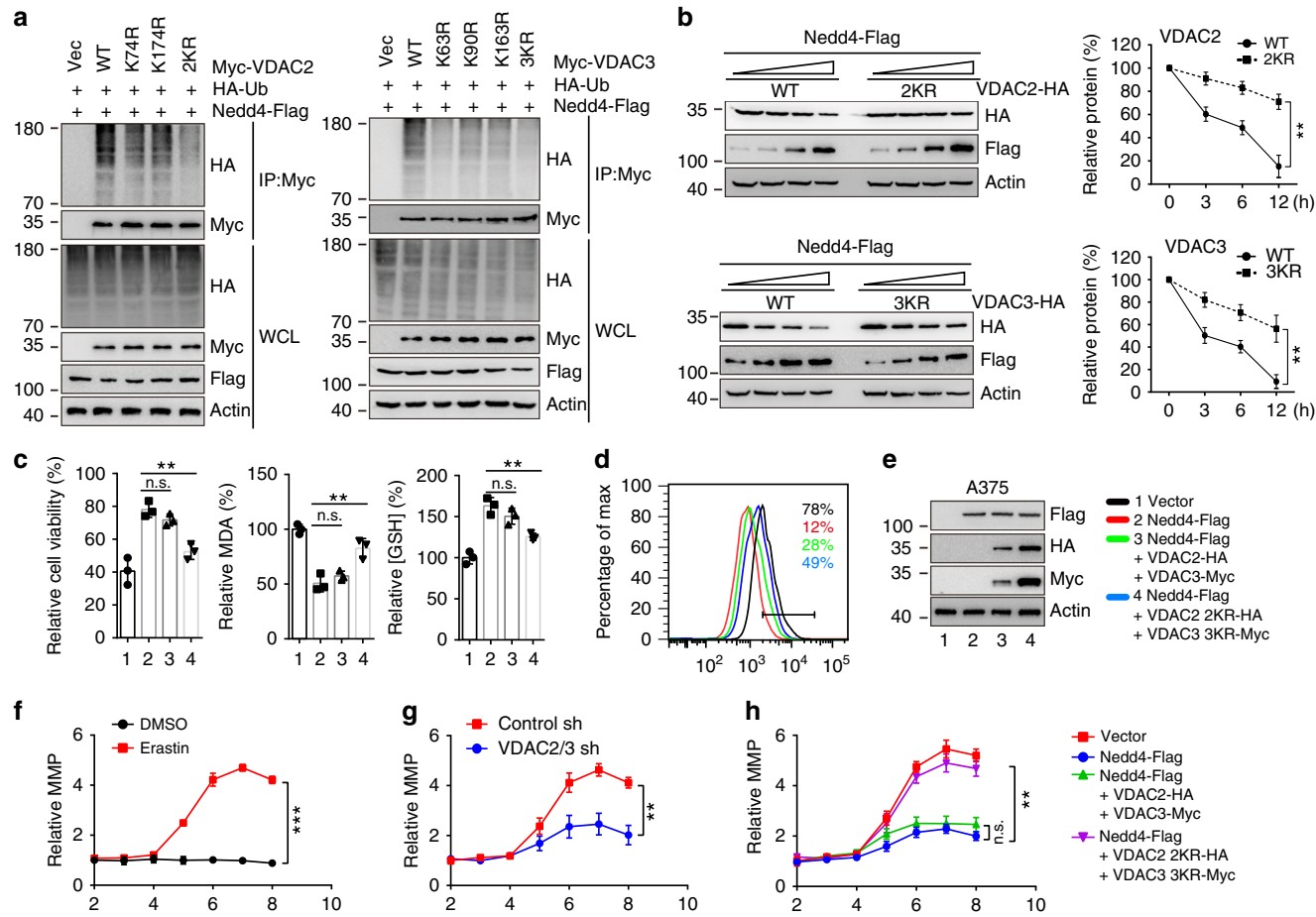

**Fig. 5 Nedd4 regulates VDAC2/3 degradation through specific ubiquitination sites. a** A375 cells were transfected with combinations of DNA constructs as indicated and treated with MG132 for 4 h. Lysates were immunoprecipitated with anti-Myc, and western blots were performed to analyze the presence of indicated proteins and levels of ubiquitination. **b** A375 cells were transfected with combinations of DNA constructs as indicated and treated with erastin (5 μM) for indicated time. Cell lysates were analyzed by immunoblotting with indicated antibodies. The protein level of VDAC2/3 was quantitated by Image J and normalized to actin levels. **c–e** Overexpression of KR mutants of VDAC2/3 partially rescued the cell sensitivity to erastin, which was inhibited by Nedd4. A375 cells were transfected with indicated DNA constructs for 48 h and treated with erastin (5 μM) for 12 h. Cell viability, intracellular MDA, and GSH levels were measured (**c**). The lipid ROS level was assessed by flow cytometry using C11-BODIPY (**d**). The protein levels of overexpressed constructs were assessed by immunoblotting (**e**). **f** Quantification of MMP during erastin-induced ferroptosis. A375 cells were treated with erastin and 50 nM TMRE was added 20 min before each indicated time point. **g** Suppression of VDAC2/3 inhibited ferroptosis-associated MMP hyperpolarization. A375 cells expressing indicated shRNA constructs were treated with erastin and 50 nM TMRE was added 20 min before each indicated time point. **h** Overexpression of VDAC2/3 KR mutants rescued MMP hyperpolarization which was inhibited by Nedd4. A375 cells expressing indicated constructs were treated with erastin and 50 nM TMRE was added 20 min before each indicated time point. Fluorescence images were acquired using a microscope and mean fluorescence was calculated by Image J software. Data shown represent mean ± SD from three independent experiments. Comparisons were made using Student's *t* test. **\*\*p* < 0.01; \*\*\*p* < 0.001; n.s. not significant.

the nonconserved amino acids regulate the differences in degradation for the VDAC isoforms.

To further understand the molecular basis of how VDAC2/3 is ubiquitinated, we searched for specific sites in VDAC isoforms that respond to Nedd4 E3 ligase. Most of the K sites among the three VDAC isoforms are conserved; however, the K63 and K163 sites changed to R, and the K90 site changed to Q in VDAC1 (Supplementary Fig. 6d). These variations may hold the key in determining the association differences and effects between Nedd4 and VDAC isoforms. We generated a series of mutations that three K sites were mutated individually and in combination. As shown in ubiquitination assays, both individual and combination mutants of VDAC2/3 were resistant to being ubiquitinated by Nedd4 (Fig. 5a). Consistently, the half-lives of VDAC2²ᴷᴿ and VDAC3³ᴷᴿ mutants were significantly extended in A375 cells transfected with Nedd4-Flag (Fig. 5b) or treated with erastin (Supplementary Fig. 7a).

To confirm the essential function of K63, K90, and K163 sites, we also generated a series of R to K mutations of VDAC1. As expected, overexpression of Nedd4 significantly increased the ubiquitin conjugation of both individual and combination RK mutants of VDAC1 (Supplementary Fig. 7b). Also, the degradation of VDAC1³ᴿᴷ was significantly increased in A375 cells transfected with Nedd4-Flag (Supplementary Fig. 7c) or treated with erastin (Supplementary Fig. 7d). To determine the biological function of the three K sites, we cotransfected WT or KR mutants of VDAC2/3 with Nedd4 shRNA into A375 cells. As shown in Fig. 5c–e and Supplementary Fig. 8a–c, overexpression of KR mutants partially rescued erastin-induced ferroptotic cell death, which was suppressed by Nedd4 overexpression. In contrast, depletion of VDAC2/3 inhibited erastin-induced ferroptosis, which was enhanced upon Nedd4 suppression in both A375 and G-361 cells (Supplementary Fig. 8e–h). These results suggest that VDAC2/3 is an essential target of Nedd4 in erastin-induced

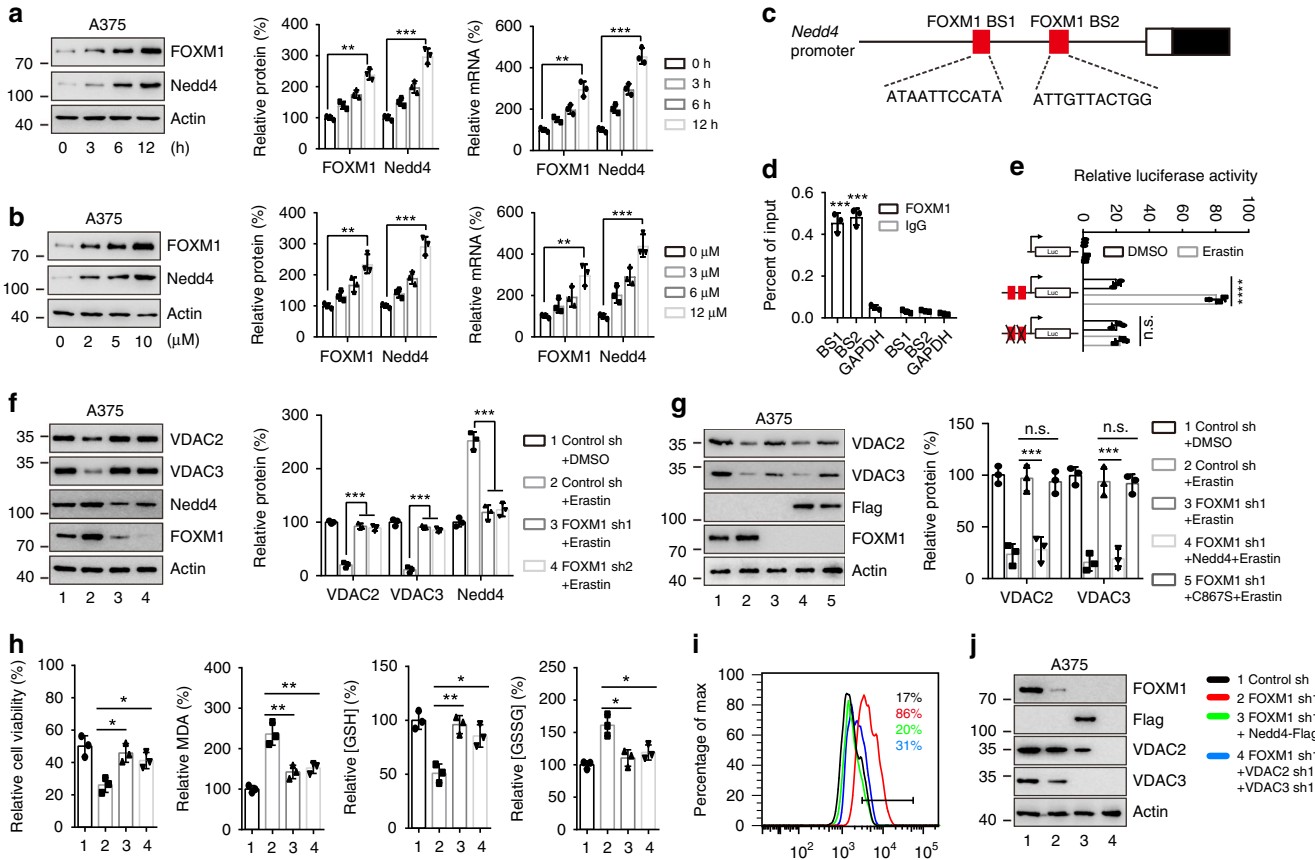

**Fig. 6 Upregulation of Nedd4 by erastin was mediated by FOXM1. a**, **b** Erastin induced the expression of FOXM1 and Nedd4. A375 cells were treated with erastin at different time points (0–12 h) or different concentrations (0–10 μM). Cell lysates were analyzed by immunoblotting with indicated antibodies. The protein levels of FOXM1 and Nedd4 were quantitated by Image J. **c** Structure of *Nedd4* promoter showing the location of two FOXM1 binding sites (in red). **d** ChIP analysis shows occupancy by FOXM1 on *Nedd4* promoter, as indicated, in A375 cells treated with erastin (5 μM) for 8 h. GAPDH serves as a negative control. **e** Dual-luciferase reporter assay showing the activity of *Nedd4* promoter in response to erastin in A375 cells. The *Nedd4* promoter reporters were transfected into A375 cells for 24 h, treated by erastin (5 μM) for an additional 12 h, then luciferase activity was measured. **f** Knockdown of FOXM1 suppressed erastin-induced Nedd4 expression and blocked VDAC2/3 degradation. A375 cells expressing control shRNA or FOXM1 shRNA constructs were treated with DMSO or erastin (5 μM) for 12 h, the protein levels of VDAC2, VDAC3, and Nedd4 were assayed by western blot. **g** Overexpression of Nedd4 but not the C867S mutant rescues the protein degradation of VDAC2/3 inhibited by FOMX1 knockdown. Cells expressing indicated DNA constructs were treated with DMSO or erastin (5 μM) for 12 h, the protein level of VDAC2/3 was assayed by western blot. **h–j** Overexpression of Nedd4 or knockdown of VDAC2/3 increased erastin-induced ferroptotic cell death which was inhibited by FOXM1 knockdown. Cells were transfected with indicated DNA constructs for 48 h, and treated with erastin (5 μM) for 12 h. Cell viability, intracellular MDA, GSH, and GSSG levels were measured (**h**). The lipid ROS level was assessed by flow cytometry using C11-BODIPY (**i**). The protein levels of indicated genes were assessed by immunoblotting (**j**). Data shown represent as mean ± SD from three independent experiments. Comparisons were made using Student's *t* test. *$p < 0.05$; **$p < 0.01$; ***$p < 0.001$; ****$p < 0.0001$; n.s. not significant.

ferroptosis. The mitochondrion plays a crucial and proactive role in erastin-induced ferroptosis but not in GPX4 inhibition-induced ferroptosis[14]. Erastin hyperpolarizes the mitochondrial membrane potential (MMP) in A375 cells (Fig. 5f). VDAC2/3 is a crucial regulatory mitochondrial membrane protein for the homeostasis of different ions and metabolites. Knockdown of VDAC2/3 affected mitochondrial function and suppressed erastin-induced MMP hyperpolarization (Fig. 5g). Besides, over-expression of Nedd4 suppressed erastin-induced MMP hyperpolarization, while overexpression of VDAC2/3 KR mutants partially rescued the inhibitory effect of Nedd4 (Fig. 5h). Taken together, our findings demonstrate that these three K sites in VDAC isoforms are essential for Nedd4-mediated ubiquitination.

**Erastin stimulates Nedd4 and FOXM1 expression**. When detecting the formation of Nedd4-VDAC2/3 complex, we observed that the protein level of Nedd4 was increased (Fig. 2b). To confirm

this phenotype, we treated A375 and G-361 cells with erastin and detected the mRNA and protein levels of Nedd4. As shown in Fig. 6a and Supplementary Fig. 9a, both the mRNA and protein levels of Nedd4 increased in a time-dependent manner. Similarly, Nedd4 levels increased with higher concentrations of erastin (Fig. 6b and Supplementary Fig. 9b). These data suggest that era-stin stimulates Nedd4 expression by transcriptional regulation.

FOXM1 is essential in the regulation of oxidative stress, and its disruption contributes to malignant transformation and tumor cell survival[15]. A previous study has demonstrated that FOXM1 upregulated Nedd4 to degrade PTEN in astrocytes by binding to specific sequences of Nedd4 promoter (Fig. 6c)[16]. Here, we found that overexpression of FOXM1 induced the expression of Nedd4 at both the mRNA and protein levels (Supplementary Fig. 9c). However, knockdown of FOXM1 by two distinct shRNA constructs did not suppress Nedd4 expression in A375 cells (Supplementary Fig. 9d), implying that FOXM1 only mediates the erastin-induced Nedd4 expression but not basal

level expression. As oncogene-induced ROS accumulation stimulates FOXM1 expression[15] and erastin accelerates ROS accumulation by directly inhibiting cystine/glutamate antiporter system $X_c^-$ activity during ferroptosis[1], we next investigated whether erastin could promote FOXM1 expression by increasing ROS. Similar to Nedd4, the expression of FOXM1 was significantly increased in A375 (Fig. 6a, b) and G-361 (Supplementary Fig. 9a, b) cells treated with erastin in a time-dependent manner.

**Erastin stimulates Nedd4 expression through inducing FOXM1.** To test whether Nedd4 is a FOXM1-responsive gene in erastin-induced ferroptosis, we performed ChIP assays and found FOXM1 directly bound to the promoter of Nedd4 in A375 cells after erastin treatment (Fig. 6c, d), but not in DMSO treated cells (Supplementary Fig. 9e). These results were consistent with the observation that FOXM1 did not regulate basal expression of Nedd4 (Supplementary Fig. 9d), further indicating that FOXM1 only mediates the erastin-induced expression of Nedd4. To determine whether FOXM1 plays a role in the activity of the Nedd4 promoter, we performed luciferase reporter assay. Both ectopic expression of FOXM1 (Supplementary Fig. 9f) and erastin treatment (Fig. 6e) drastically induced the activity of Nedd4 promoter, but not the FOXM1 binding sites mutant. Conversely, inactivation of endogenous FOXM1 by two distinct shRNA abolished the erastin-induced transcriptional activation of Nedd4 promoter, but not the basal transcriptional activity (Supplementary Fig. 9g). These results are consistent with the ChIP assay results and confirm that FOXM1 binds to the Nedd4 promoter to induce expression in response to erastin treatment.

To further confirm that erastin-induced expression of Nedd4 is regulated by FOXM1, we generated two stable cell lines expressing different FOXM1 shRNA constructs. The induction of Nedd4 was abrogated when FOXM1 was depleted in erastin-stimulated cells, with a consequently elevated protein level of VDAC2/3 (Fig. 6f). These results suggest that erastin regulates Nedd4 via FOXM1. Moreover, the induction of VDAC2/3 by knocking down FOXM1 was absent in cells overexpressing Nedd4 but not Nedd4$^{C867S}$ (Fig. 6g), suggesting that FOXM1 regulates VDAC2/3 through Nedd4. Interestingly, inactivation of oncogenic RAS-RAF signaling by knocking down BRAF (Supplementary Fig. 9h) or antioxidant treatment (Supplementary Fig. 9i) abolished erastin-induced expression of FOXM1 and Nedd4, and as a result, increased the expression of VDAC2/3. Thus, both oncogenic RAS-RAF signaling and ROS play essential roles in inducting FOXM1 and Nedd4 in A375 cells.

We further validated that FOXM1-Nedd4-VDAC2/3 pathway regulated ferroptosis in melanoma cells. Knockdown of FOXM1 increased erastin-induced ferroptosis in both A375 and G-361 cells (Fig. 6h–j and Supplementary Fig. 9j–l). Overexpression of Nedd4 in FOXM1 depleted cells partially suppressed erastin-induced ferroptosis (Fig. 6h–j and Supplementary Fig. 9j–l). Similarly, knockdown of VDAC2/3 in FOXM1 depleted cells also partially inhibited erastin-induced ferroptosis (Fig. 6h–j and Supplementary Fig. 9j–l). Furthermore, overexpression of KR mutants of VDAC2/3 increased the sensitivity of cells to erastin, which was suppressed by FOXM1 overexpression (Supplementary Fig. 10a–f). Collectively, these findings suggest that FOXM1 inhibits ferroptosis by regulating Nedd4 expression and subsequent VDAC2/3 degradation in melanoma cells.

**Inhibition of Nedd4 enhances erastin sensitivity.** The observed effects of Nedd4 on the protein degradation of VDAC2/3 suggest that the expression level of Nedd4 should modulate erastin sensitivity in melanoma cells. Thus, we stably knocked down Nedd4 in A375 and G-361 cells by lentiviral vector following erastin

treatment and measured its effect on cell viability by colony formation assay. We found that Nedd4 depletion significantly enhanced erastin-induced ferroptotic cell death in both A375 and G-361 cells (Fig. 7a, b). To determine whether suppression of Nedd4 enhances the anti-cancer activity of erastin in vivo, Nedd4 knockdown A375 cells were implanted into the subcutaneous space of nude mice. When the tumors reached 50 mm³, mice were treated with erastin for 20 days. Compared with the control vector group, repression of Nedd4 reduced the size of tumors formed (Fig. 7c) and exhibited increased MDA levels and reduced GSH levels (Fig. 7d). Besides, we performed 4-hydroxy-2-noneal (4HNE) immunohistochemistry (IHC) analysis to characterize lipid peroxidation levels in tumor xenograft samples. The staining of 4HNE was significantly increased after erastin treatment in both control cells and Nedd4 depletion cells (Fig. 7e), suggesting that erastin increased the lipid peroxidation level of tumor xenograft samples. Consistently, the staining of 4HNE in Nedd4 depletion cells was stronger than control cells (Fig. 7e), which further confirmed that suppression of Nedd4 enhanced erastin-induced ferroptosis in vivo.

To confirm the essential function of FOXM1 and VDAC2/3 in erastin-induced ferroptosis, we generated A375 stable cell lines expressing FOXM1 shRNA construct or VDAC2/3 overexpression construct. Knockdown of FOXM1 or overexpression of VDAC2/3 enhanced the antitumor activity of erastin in both colony formation assay and mice xenograft model (Supplementary Fig. 11a–d). To test whether Nedd4-mediated VDAC2/3 degradation is also essential for the suppression of erastin-induced ferroptosis in melanoma cells carrying wt BRAF, we knocked down Nedd4 or overexpressed VDAC2/3 in MeWo cells. Compared with the control groups, repression of Nedd4 or overexpression of VDAC2/3 sharply reduced the size of tumors formed in a mice xenograft model (Supplementary Fig. 11e, f). In addition to BRAF mutants, there are other subtypes of melanoma that carry other mutations, such as NRAS (SK-MEL-2 and WM2032 cells), TP53 (SK-MEL-3 cells), and PTEN (SK-MEL-24 cells). To further verify whether Nedd4 and VDAC2/3 play essential roles in erastin-induced ferroptosis, we knocked down Nedd4 or overexpressed VDAC2/3 in these cells bearing NRAS, TP53 or PTEN mutations. Consistently, suppression of Nedd4 or overexpression of VDAC2/3 enhanced the antitumor activity of erastin in mice (Supplementary Figs. 12a–d and 13a–d). Taken together, these results support the idea that Nedd4 and VDAC2/3 play essential roles in regulating erastin-induced ferroptosis in different subtypes of melanoma cells.

## Discussion

Nedd4 has a vital role in many physiological processes. Several studies have demonstrated that Nedd4 can act as an oncogene in promoting cancer cell growth. Overexpression of Nedd4 facilitated tumor cell growth, whereas depletion of Nedd4 significantly inhibited the proliferation of cancer cells in vitro and tumor growth in vivo[17–19]. As an E3 ligase, Nedd4 mainly exerts its oncogenic function mediating the degradation of PTEN in an ubiquitination-dependent manner[18,20]. Besides PTEN, various studies also revealed a critical role for Nedd4 in tumorigenesis by regulating other substrates, including ENaC (Epithelial sodium channel)[21], Notch[22,23], pAKT[24], WW45[25], LATS1 (large tumor suppressor kinase 1)[26], and many others. Here we identified VDAC2 and VDAC3 as two substrates of Nedd4 during erastin-induced ferroptosis in melanoma cells (Fig. 7f). Knockdown of Nedd4 leads to elevated protein level of VDAC2/3, which increased the sensitivity of melanoma cells to erastin both in vitro and in vivo. Overall, abnormal levels of Nedd4 appear to consistently exert oncogenic tendencies through various regulatory

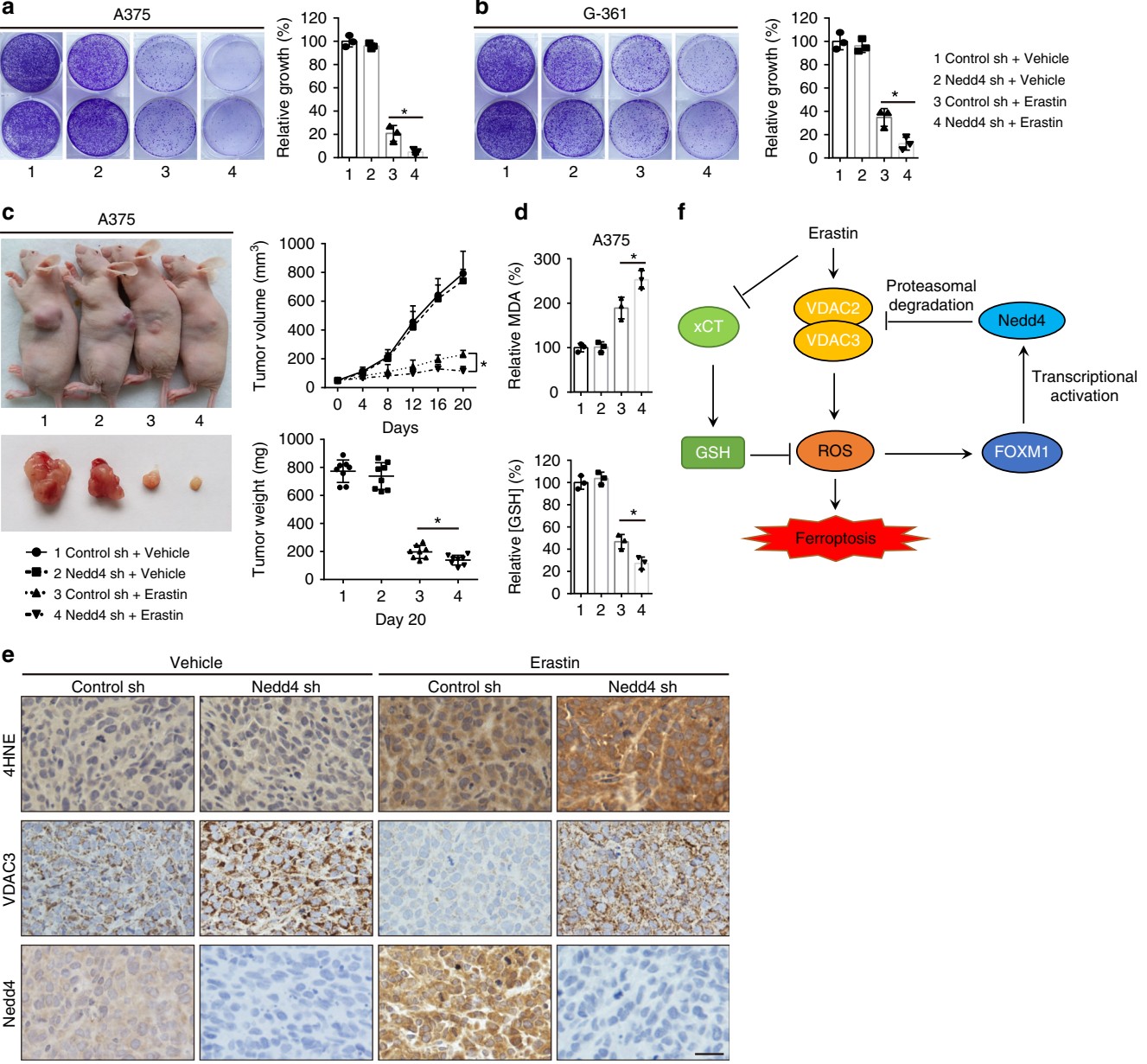

**Fig. 7 Knockdown of Nedd4 suppresses erastin resistance in vitro and in vivo. a, b** Colony formation assay of indicated A375 (**a**) and G-361 (**b**) cells. Cells were treated with erastin (5 μM) for 24 h and were grown without erastin for 10 days. For each cell line, all dishes were fixed, stained, and photographed at the same time. **c, d** Knockdown of Nedd4 enhanced erastin-induced ferroptosis in vivo. The 7-week-old immunodeficient nude mice (eight mice per group) were injected subcutaneously with indicated A375 cells ($5 \times 10^6$ cells per mouse) and treated with erastin (15 mg per kg intraperitoneal, twice every other day) when the tumor volume reached 50 mm³. Tumor volume was calculated every 4 days (**c**), and the tumor mess was measured at day 20 (**c**). The relative levels of MDA and GSH of indicated tumors were measured (**d**). **e** Immunohistochemistry analysis of VDAC3, Nedd4, and 4HNE of tumor xenografts. Scale bar, 50 μm. The experiment was repeated twice independently with similar results. **f** Schematic depicting the regulation of VDAC2/3 by FOXM1 and Nedd4 during ferroptosis in melanoma cells. Erastin activates FOXM1 to induce the transcription of Nedd4. Increased level of Nedd4 degrades VDAC2/3 and suppresses ferroptotic cell death. Data are mean ± SD from three independent experiments. Comparisons were made using Student's *t* test. *$p < 0.05$.

substrates across several human cancers. In this regard, targeting Nedd4 could be a promising therapeutic strategy for the treatment of human malignancies.

A previous report shows that erastin directly binds to VDAC2 using a radiolabelled analog and a filter-binding assay[10]. Meanwhile, erastin downregulates the protein level of VDAC2/3[10], which is consistent in our experiment. Whether erastin-induced VDAC2/3 degradation is related to erastin-VDAC2 interaction is an interesting and important question. Unfortunately, the amino acids of VDAC2 which mediate erastin interaction have not been

identified yet. It is of considerable significance to detect whether a VDAC2 mutant that lost the erastin binding ability can still be degraded by erastin. Based on our data, it is possible that erastin degrades VDAC2/3 through ROS-induced expression of FOXM1 and Nedd4, without necessarily binding to it. One testament is that RSL3, which does not bind to VDAC2, still promotes the degradation of VDAC2/3 through FOXM1 and Nedd4. However, knockdown of VDAC2/3 suppressed the sensitivity of cell to erastin but not RSL3. A recent study showed that the mitochondrion plays a crucial and proactive role in cysteine-

deprivation-induced ferroptosis but not in GPX4 inhibition-induced ferroptosis[14]. VDAC2/3 is expressed in the external membrane of the mitochondrion and regulates the movement of numerous ions and metabolites between the cytosol and mitochondrion. Loss of VDAC2/3 may affect mitochondrial activity by disrupting the homeostasis of different ions and metabolites. This hypothesis is corroborated by the suppression of erastin-induced MMP hyperpolarization by VDAC2/3 knockdown. All these results suggest that the function of VDACs in ferroptosis is closely related to their roles within mitochondria.

It is not surprising that the expression level of Nedd4 is precisely regulated. A previous study showed that Nedd4 is upregulated in response to hydrogen superoxide through transcriptional activation likely mediated by ROS-responsive FOXM1[27]. In our study, we also found that erastin-induced ROS activates FOXM1, which then binds to the promoter region of Nedd4 to increase the transcription level. FOXM1 is a member of the Forkhead box (Fox) transcription factor family and has critical functions in tumor development and progression. Here, we identify a negative feedback loop involving FOXM1 that regulates erastin-induced ferroptosis. We showed that induction of FOXM1 by erastin treatment requires ROS. Elevated FOXM1, in turn, downregulates ROS level by transcriptional activation of Nedd4 to degrade VDAC2/3. Intriguingly, overexpression of Nedd4 could only partially rescue the reduced cell viability mediated by FOXM1 knockdown. This phenotype might be explained by the finding that FOXM1 also stimulates the expression of ROS scavenger genes, such as MnSOD, catalase, and PRDX3[15]. In response to ROS, FOXM1 simultaneously promote transcription of Nedd4 and ROS scavenger genes to limit the accumulation of ROS. Thus, besides the development of Nedd4 inhibitors, it might be an alternative strategy to target the upstream effector, FOXM1, in order to govern the expression levels of Nedd4 and ROS scavenger genes simultaneously.

In mammals, all three isoforms of VDAC (VDAC1, VDAC2, and VDAC3) have been identified, characterized, and sequenced. All of them operate at the external membrane of the mitochondrion. Although they have very similar structures, these three isoforms have different functions that do not necessarily compensate for the failings of one or the other. VDAC1 knockout mice show partial lethality phenotype during embryogenesis. VDAC2 knockout mice die in the embryonic stage and cannot be rescued by overexpression of VDAC1. VDAC3 does not seem essential for cell survival[9]. Here we found that Nedd4 mainly degrades VDAC2 and VDAC3 but not VDAC1 during erastin-induced ferroptosis, although all three isoforms have PPxY motifs that bind to the WW domain of Nedd4. By comparing the sequences of three VDAC isoforms, we found that there are three K sites of VDAC1 different from VDAC2/3. The ubiquitination of R63K, Q90K, and R163K mutants of VDAC1 are slightly increased when compared with wild-type VDAC1, suggesting these sites mediate the stability of VDAC1. Intriguingly, these three isoforms have different expression patterns. For example, low expression of VDAC1 was observed in the frontal cortex and thalamus in patients with Alzheimer's disease, whereas high expression of VDAC2 was described in the temporal cortex[28]. The differing expression patterns and regulatory mechanisms among the three isoforms expose VDAC as an interesting variable in differentiating disease mechanisms.

A significant issue accompanied by treating cancer cells is resistance to therapeutics. The mechanism of drug resistance is complicated and remains incompletely understood. Luckily, recent studies demonstrate that cancer cells existing in drug-resistant state exhibit exquisite sensitivity to ferroptotic cell death[3–5]. As the first identified ferroptosis inducer, erastin is an effective drug to induce cell death in a variety of tumors[5–7,29]. Unfortunately, the efficacy of erastin is suppressed by its mechanism, namely the degradation of VDAC2/3. This negative feedback process limits the practical usage of erastin as an ideal drug to treat tumors effectively. Whether this type of resistance also occurs in other ferroptosis inducers is still unknown. However, a great lesson can be observed from erastin as we continue to understand how cells respond to drugs and manifest resistance. The cellular response bringing about resistance to erastin is quite profound, and we must understand how and why this cellular disruption is well defended, whereas other drugs cause minor retaliation by the cell that does not manifest as drug resistance. We have laid the groundwork in determining the mechanism of how the cell reacts to erastin, and we can use this framework to understand how adjunct therapy can be used and to develop strategies involving ferroptotic players to destroy previous drug-resistant cancer cells.

## Methods

**Cell culture and transfection**. All cell lines were purchased from the American Type Culture Collection (ATCC) and maintained at 37 °C with 5% CO2. A375 (ATCC®CRL-1619), G-361 G361 (ATCC®CRL-1424), MeWo (ATCC®HTB-65), SK-MEL-2 (ATCC®HTB-68), SK-MEL-3 (ATCC®HTB-69), SK-MEL-24 (ATCC®HTB-71), WM2032 (ATCC®HTB-70), and HEK293T (ATCC®CRL-3216) cells were cultured in Dulbecco's modified Eagle's medium supplemented with 10% fetal bovine serum (FBS; Invitrogen), 2 mM L-glutamine, and 1% penicillin–streptomycin (Gibco-BRL). Transfections were performed according to the manufacturer's instructions with lipofectamine 2000 (Invitrogen) or calcium phosphate transfection kit (Thermo). None of the cell lines used in this study was found in the database of commonly misidentified cell lines that is maintained by ICLAC and NCBI Biosample. All cell lines were tested and confirmed to be free of mycoplasma.

**Antibodies and chemicals**. The following antibodies were used in this study at the indicated dilution for western blot (WB) analysis, IP, IHC, and immuno-fluorescence (IF): Nedd4 (PA5-17463, Thermo, 1:1000 for WB, 1:100 for IP, 1:100 for IHC), VDAC1 (MABN504, Merck, 1:1000 for WB), VDAC2 (ab37985, Abcam, 1:1000 for WB), VDAC3 (ab130561, Abcam, 1:1000 for WB, 1:100 for IHC), FOXM1 (702664, Thermo, 1:1000 for WB), p-ERK (4370, Cell Signaling, 1:1000 for WB), ERK (4695, Cell Signaling, 1:1000 for WB), 4HNE (ab46545, Abcam, 1:100 for IHC), Flag (F3165, clone M2; Sigma, 1:1000 for WB, 1:100 for IP), HA (H6533, Sigma, 1:1000 for WB), GST (PA1982A, Thermo, 1:1000 for WB), Myc (AH00052, Thermo, 1:1000 for WB, 1:100 for IP), and Actin (PA116889, Thermo, 1:1000 for WB). Horseradish peroxidase (HRP)-labeled or fluorescently labeled secondary antibody conjugates were purchased from Molecular Probes (Thermo). Purified rabbit IgG was purchased from Pierce. Erastin (E7781), Ferrostatin-1 (S7243), Liproxstatin-1 (S7699), EUK134 (S4261), and Tempol (S2910) were obtained from Selleck (Houston, TX, USA). DFO (D9533) and CPX (SML2011) were obtained from Sigma (St. Louis, MO).

**Plasmids**. The Flag- or HA-tagged wild-type (wt) Nedd4 and Nedd4^ΔC2, Nedd4^ΔWW, and Nedd4^ΔHECT mutants were constructed by cloning the cDNA of the full-length or truncated mutants into the KpnI and XhoI sites of the pcDNA5/FRT/TO vector. pCDH/puro-Flag-Nedd4 wt and mutants were constructed by cloning the cDNA of the WT or truncated Nedd4 mutants into the EcoRI and XhoI sites of the pCDH-CMV-MCS-EF1-Puro vector. The Nedd4 mutants C867S and shRNA-resistant Nedd4 were generated using the Q5 Site-Directed Mutagenesis Kit (NEB) and then cloned in frame with a Flag tag into pcDNA5/FRT/TO or pBabe-neo vectors. The full-length cDNA of human VDAC1, VDAC2, VDAC3, and FOXM1 were cloned from HEK293T cDNA by PCR. The Flag-, Myc- or HA-tagged WT VDAC1, VDAC2, and VDAC3, and their mutant derivatives were constructed by cloning the cDNA of the full-length or truncated mutants into the KpnI/NotI sites of the pcDNA5/FRT/TO vector. The Flag-, Myc-, or HA-tagged FOXM1 were constructed by cloning the cDNA into the AflII and XhoI sites of the pcDNA5/FRT/TO vector. All shRNAs were purchased from Sigma. The sequence targeting Nedd4 is: 5′-TGCAAGCACAACGTGCATTTA-3′ and 5′-TACGTGAG AGTGACGTTATAT-3′; targeting VDAC2 is: 5′-CACTGCTTCCATTTCTGC AAA-3′; targeting VDAC3 is: 5′-GCAACCTAGAAACCAAATATA-3′; targeting FOXM1 is: 5′-TTGCAGGGTGGTCCGTGTAAA-3′ and 5′-AGGACCACTT TCCCTACTTTA-3′; targeting BRAF is: 5′-TTGCTGGTGTATTCTTCATAG-3′, and 5′-TTTGAAGGCTTGTAACTGCTG-3′. All constructs were confirmed by sequencing.

**Cell viability assay**. Cell viability was evaluated using the cell counting kit-8 (CCK-8) (#96992, Sigma) according to the manufacturer's instructions. Cells were seeded in 96-well plates (10,000 cells per well) and treated with DMSO, erastin or

RSL3 for 24 h. The culture medium was replaced with 100 μl fresh medium containing 10 μl of the CCK-8 solution for each well of the plate. And the culture was returned to the cell culture incubator for 1–4 h. Measure the absorbance at 450 nm using a microplate reader. This assay uses the highly water-soluble tetrazolium salt WST-8 [2-(2-methoxy-4-nitrophenyl)-3-(4-nitrophenyl)-5-(2,4-disulfophenyl)-2H-tetrazolium, monosodium salt] to produce a water-soluble formazan dye upon reduction in the presence of an electron carrier. Absorbance at 450 nm is proportional to the number of living cells in the culture.

**Malondialdehyde (MDA) assay**. The relative MDA concentration in cell lysates was assessed using a lipid peroxidation assay kit (ab118970) purchased from Abcam according to the manufacturer's instructions. Cells were seeded in 10 cm plate ($5 \times 10^6$ cells per plate) and treated with DMSO, erastin or RSL3 for 24 h. Cells or 10 mg tumor tissue were washed with ice-cold PBS and homogenized on ice in 300 μl of the MDA lysis buffer with 3 μl BHT (100×), then centrifuged ($13,000 \times g$, 10 min) to remove insoluble material. Place 200 μl of the supernatant from each homogenized sample into a microcentrifuge tube. Add 600 μL of the thiobarbituric acid (TBA) solution into each vial. Incubate samples at 95 °C for 60 min. The MDA in the sample reacted with TBA to generate an MDA–TBA adduct. Cool samples to room temperature in an ice bath for 10 min. Pipette 200 mL from each reaction mixture into a 96-well plate for analysis. Measure the absorbance at 532 nm using a microplate reader.

**Iron assay**. Intracellular ferrous iron ($Fe^{2+}$) level was determined using the iron assay kit (ab83366) purchased from Abcam according to the manufacturer's instructions. Cells were seeded onto 10 cm plate ($5 \times 10^6$ cells per plate) and treated with DMSO, erastin or RSL3 for 24 h. Cells were collected and washed in ice-cold PBS and homogenized in 5× volumes of iron assay buffer on ice, then centrifuged ($13,000 \times g$, 10 min) at 4 °C to remove insoluble material. Collect the supernatant and add iron reducer to each sample, mix, and incubate for 30 min at room temperature. Add 100 μl of iron probe to each sample. Mix well using a horizontal shaker or by pipetting, and incubate the reaction for 60 min at room temperature. Protect the plate from light during the incubation. Measure the absorbance at 593 nm using a microplate reader.

**Lipid ROS assay using flow cytometer**. Lipid ROS level was analyzed by flow cytometry using BODIPY-C11 dye[1,8]. Cells were seeded at a density of $2.5 \times 10^5$ per well in six-well plates and grown overnight. Cells were treated with DMSO or erastin for 24 h. The culture medium was replaced with 2 ml medium containing 5 μM of BODIPY-C11 (Thermo Fisher, Cat# D3861), and cells were returned to the cell culture incubator for 20 min. Cells were harvested in 15 ml tubes and washed twice with PBS to remove excess BODIPY-C11, cells then were resuspended in 500 μl of PBS. The cell suspension was filtered through cell strainer (0.4 μm nylon mesh) and subjected to the flow cytometry analysis to examine the amount of lipid ROS within cells. Oxidation of BODIPY-C11 resulted in a shift of the fluorescence emission peak from 590 nm to 510 nm proportional to lipid ROS generation. The fluorescence intensities of cells per sample were determined by flow cytometry using the BD FACS Aria cytometer (BD Biosciences). A minimum of 10,000 cells was analyzed per condition. The gating strategy for the flow cytometry is provided in the Source Data file.

**Glutathione assay**. The relative glutathione (GSH) concentration in cell or tissue lysates was assessed using the glutathione assay kit (CS0260; Sigma) according to the manufacturer's instructions. Cells were seeded onto 10 cm plate ($5 \times 10^6$ cells per plate) and treated with DMSO, or erastin for 24 h. Cells were washed with ice-cold PBS and lysed in 1% lysis buffer (25 mM Tris at pH 7.5, 300 mM NaCl, 1 mM EDTA, 0.5% NP-40) supplemented with a phosphatase inhibitor mix (Pierce) and a complete protease inhibitor cocktail (Roche). After sonication (Thermo Model 120, amplitude 15%, process time 10 s, push-on time 5 s, and push-off time 1 s), cell lysates were centrifuged at 13,200 rpm at 4 °C for 10 min, and cleared lysate was used to determine the amount of GSH in the sample. Enzymes were removed by using deproteinizing sample kit (ab204708), which can interfere with the analysis. Add 50 μL of GSH assay mixture into each sample well and incubate it at room temperature for 60 min protected from light. The measurement of GSH used a kinetic assay in which catalytic amounts (nmoles) of GSH caused a continuous reduction of 5,5′-dithiobis (2-nitrobenzoic acid) to 5-thio-2-nitrobenzoic acid. The reaction rate was proportional to the concentration of glutathione up to 2 mM. The yellow product (5-thio-2-nitrobenzoic acid) was measured spectrophotometrically at 412 nm.

**GSSG assay**. The relative GSSG concentration in cell lysates was assessed using a kit from Cayman (#703002) according to the manufacturer's instructions. Cells were seeded onto 10 cm plate ($5 \times 10^6$ cells per plate) and treated with DMSO or erastin for 24 h. Cells were washed with ice-cold PBS and collected by a rubber policeman. Cells were homogenized in 2 ml of cold 50 mM MES buffer, and then centrifuged ($10,000 \times g$, 15 min) at 4 °C to remove insoluble material. The quantification of GSSG, exclusive of GSH, was accomplished by derivatizing GSH with 2-vinylpyridine (Sigma, 13229-2). Collect the supernatant and add 10 μl of 2-vinylpyridine solution (1 M in ethanol) per ml of sample. Mix well on a vortex and

incubate for 60 min at room temperature to block the thiol group of the GSH already present. NADPH (95 μl of 2 mg per ml) in water and 5 μl of 2 U/ml glutathione reductase were added to reduce GSSG. Add 50 μl of sample per well of 96-well plate. Prepare the assay cocktail mixture and add 150 μl to each wells containing samples. Incubate the plate in the dark on a shaker for 30 min. Measure the absorbance at 412 nm using a microplate reader.

**Measurement of mitochondrial membrane potential**. Cells were incubated with 50 nM TMRE (T669, Thermo) for 20 min, then extensively washed with PBS twice to remove traces of TMRE from the media. Fluorescence images were acquired using a Zeiss Axiolmager M2 microscope fitted with a 60× objective and Zeiss image software. The experimenters were blind to the sample identity during analyses. Approximately 100 cells from 10-20 randomly chosen high-power fields (×60) were evaluated for imaging analysis. Values indicate the mean ± SD of at least three independent experiments.

**Immunohistochemistry**. Tissue sections from the indicated mouse models were fixed in 10% buffered formalin and embedded in paraffin. For IHC staining, tissue slides were deparaffinized in xylene and rehydrated in alcohol. Endogenous peroxidase was blocked with 3% hydrogen peroxide. Antigen retrieval was achieved using a microwave and 0.1 M citric sodium buffer (pH 6.0). Sections were then incubated overnight at 4 °C with the primary antibody. Antibody binding was detected with HRP-DAB kit (ZLI-9019, ZsBio). Sections were then counterstained with haematoxylin. For negative control, the primary antibody was replaced with the buffer. Images were acquired using a Nikon microscope and Nikon image software.

**Immunofluorescence microscopy**. Cells plated on coverslips were fixed with 4% paraformaldehyde (20 min at room temperature (RT)). After fixation, cells were permeabilized with 0.2% Triton X-100 for 10 min and blocked with 10% goat serum (Gibco) for 1 h. The expression of VDAC1-mcherry, VDAC2-GFP, and VDAC3-GFP was detected by autofluorescence without primary antibody staining. Cells were then extensively washed with PBS buffer and incubated with DAPI (4′, 6′-diamidino-2-phenylindole) in 1% goat serum for 10 min at RT. Cells were mounted using Vectashield (Vector Laboratories, Inc.). Images were acquired using a Zeiss Axiolmager M2 microscope fitted with a 60× objective and Zeiss image software. For multichannel imaging, fluorescent staining was imaged sequentially in line-interlace modes to eliminate crosstalk between the channels. For image quantification, approximately 200 cells, randomly chosen from 20 high-power fields (×60) and pooled from three independent experiments, were evaluated for the distribution pattern of the indicated molecules. All experiments were independently repeated several times. The experimenters were blind to the sample identity during analyses.

**Immunoprecipitation and immunoblotting**. For IP, cells were washed with ice-cold PBS and lysed in 1% NP40 lysis buffer (25 mM Tris at pH 7.5, 300 mM NaCl, 1 mM EDTA, and 1% NP40) supplemented with a complete protease inhibitor cocktail (Roche). The lysates were sonicated (Thermo Model 120, amplitude 15%, process time 10 s, push-on time 5 s, and push-off time 1 s), and then were rotated for 1 h at 4 °C. The cell lysates were centrifuged at $20,000 \times g$ for 10 min at 4 °C to collect soluble fraction. The protein A/G agarose beads were added to the samples for preclearing at 4 °C for 1 h. The samples were centrifuged at $2000 \times g$ for 2 min at 4 °C, and the whole-cell lysates (WCLs) were transferred to new sterile tubes for IP with the indicated antibodies. Typically, 1–4 μg commercial antibody was added to 1 ml WCLs, followed by incubation for 8–12 h at 4 °C. Fresh protein A/G agarose beads were added and incubation of the samples was continued for another 2–4 h. Immunoprecipitates were extensively washed with IP wash buffer (10 mM Tris at pH 7.5, 150 mM NaCl, 1 mM EDTA, and 0.2% Triton X-100) supplemented with 1× protease inhibitor mix (Roche), and then eluted with 100 μl of sodium dodecyl sulfate-polyacrylamide gel electrophoresis (SDS-PAGE) loading buffer by boiling for 5 min. The eluates were then subjected to WB with the indicated antibodies. For immunoblotting, cells were trypsinized and washed twice with ice-cold PBS, lysed in lysis buffer (20 mM Tris at pH 7.5, 150 mM NaCl, 1 mM EDTA, and 2% Triton X-100) supplemented with a complete protease inhibitor cocktail (Roche). The SDS-PAGE loading buffer was added to cell lysates and boiled for 5 min. Cell lysates were resolved by SDS-PAGE and transferred to polyvinylidene difluoride membranes (Bio-Rad). Membranes were blocked with 5% nonfat milk or bovine serum albumin, and detected with the indicated first antibodies and HRP-conjugated goat secondary antibodies (1:5000, Invitrogen). Immunodetection was achieved using the Hyglo chemiluminescence reagent (Thermo), and detection was performed by a GE ECL machine.

**Chromatin immunoprecipitation (ChIP) analysis**. ChIP was performed using MAGnify™ Chromatin IP System (Thermo 492024) according to the manufacture instruction. Cells were crosslinked with 1% formaldehyde for 10 min at room temperature and quenched with 0.125 M glycine for 5 min at room temperature. The crosslinked cells were collected by cell scraper and centrifuged at $200 \times g$ for 10 min at 4 °C, then washed twice with cold PBS buffer and resuspended in lysis buffer supplemented with proteinase inhibitors for 1 h at 4 °C. The lysates were

sonicated (Thermo Model 120, amplitude 30%, process time 10 s, push-on time 5 s, and push-off time 1 s) to yield 200–500 base pair DNA fragments. Samples were cleared by centrifugation at $20,000 \times g$ for 10 min at 4 ºC, the chromatin-containing supernatant was transferred to new sterile tubes. Dilute the chromatin samples in cold dilution buffer prepared with protease inhibitors to a final volume of 100 μl per reaction. Dynabeads protein A/G prepared in cold dilution buffer and antibodies were added to each tube. IP was performed by incubating diluted chromatin with the appropriate antibody-Dynabeads protein A/G complex overnight at 4 °C. The beads were washed five times each with IP buffer 1 and IP buffer 2 at 4 °C. The liquid was discarded and cross-linking buffer prepared with proteinase K was added into the IP sample tubes. The beads were fully resuspended and incubated at 55 °C for 15 min. The IP sample liquids were transferred to new sterile tubes and incubated at 65 °C for 30 min. DNA purification magnetic beads prepared with DNA purification buffer were added to each tube and incubated at room temperature for 5 min. The DNA purification magnetic beads were washed five times using DNA wash buffer. After washing, the DNA was extracted with DNA elution buffer by incubating at 55 °C for 20 min. The purified DNA was then quantified by quantitative PCR with the primer sets adjacent to the listed gene promoters. Nedd4 BS1 forward, 5′-TCTTCGTCACTGCCTTCTG-3′ and Nedd4 BS1 reverse, 5′-TT GGACTCGGGATCTGAAA-3′; Nedd4 BS2 forward, 5′-TATGGATGCCTTATT TGGTG-3′ and Nedd4 BS2 reverse, 5′-TGAACTAGGACCTCCTGTAGAAT-3′; GAPDH forward, 5′-GGCAGCACAGCCCACAGGTT-3′ and GAPDH reverse, 5′-ATCGTGACCTTCCGTGCAGAAAC-3′.

### Luciferase reporter assay.
The human Nedd4 promoter was amplified by PCR from the genomic DNA of 293 T cells, and cloned into the pGL-4.23 vector. To generate the BS1/BS2 mutant construct, we deleted the binding sequences (ATAATTCCATA and ATTGTTACTGG) using the NEB Q5 site-directed mutagenesis kit. The mutant construct was verified by sequencing. Cells were cultured in six wells plates and transfected with the promoter construct along with the pRL-CMV Renilla luciferase reporter plasmid as an internal control using polyfect transfection reagent (Qiagen 301107). Cells were lysed 48 h after transfection and assayed with the dual-luciferase reporter assay system (Promega E1910). To prepare cell lysis, remove growth media form cultured cells and wash cells in 1X PBS buffer. Add 500 μl 1× PLB buffer into each sample and gently shake for 20 min. Then, collect the PLB lysate and dispense samples into 96-well plate (20 μl per well). Add the luciferase assay substrate and measure the luciferase activity by using Beckman-Coulter DTX880. At least four replicates with three independent experiments were performed; transfection efficiency was normalized using Renilla luciferase. The PCR primers of Nedd4 promoter are Nedd4 promoter forward, 5′-TGGAGGTCTTCCT GTAATTATGGA-3′ and Nedd4 promoter reverse 5′-AGGAGGCGCTCCCTCAG CGACAGCA-3′.

### Protein purification and GST binding assay.
To identify Nedd4-VDAC2 and Nedd4-VDAC3 binding complexes, DNA fragment corresponding to Nedd4 was cloned into the pGEX-4T-1 vector. The GST and GST-Nedd4 proteins were expressed in the BL21(DE3) strain. Cells were harvested and disrupted by sonication in PBS supplemented with a complete protease inhibitor cocktail (Roche). The clear lysate was applied onto a gravity flow column containing glutathione resin (Sigma, G4510). Purified VDAC2 and VDAC3 proteins were then mixed with glutathione resin, and the binding reaction was incubated for 2 h at 4 °C. Precipitates were washed extensively with NP40 lysis buffer. Proteins bound to glutathione beads were eluted with 10 mM reduced glutathione (Sigma, G4251) in 50 mM Tris-HCl, and detected by immunoblotting with indicated antibodies.

For VDAC2 and VDAC3 purification, DNA fragments corresponding to VDAC2 and VDAC3 fused to an N-terminal $(His)_{10}$-YFP tag was cloned into a mammalian expression vector derived from the pXLG vector. The resulting plasmid was used to transfect HEK293E cells in suspension. After incubation for 3 days, the cells were harvested and disrupted by sonication in a buffer containing 100 mM NaCl, 20 mM Tris-HCl (pH 7.5), and 5 mM β-mercaptoethanol. Cleared cell lysate was loaded onto a Ni-NTA column (Qiagen), and the fusion protein was eluted with the same buffer containing an additional 200 mM imidazole. After cleavage of the protein by the PreScission Protease (GE healthcare), VDAC2 and VDAC3 proteins were further purified using a Hitrap Q anion exchange column (GE healthcare) and HiLoad 26/60 Superdex 200 gel filtration column (GE Healthcare).

### RNA extraction, cDNA synthesis, and qPCR analysis.
Total RNA was isolated with the RNeasy Mini Kit (Qiagen 74104), and 1 μg of total RNA was used for cDNA synthesis using the iScript™ cDNA Synthesis Kit (Bio-rad). Quantitative PCRs were carried out using iQ SYBR Green Master Mix (Bio-rad). Samples were obtained and analyzed on the CFX96 Touch Quantitative PCR Detection System. The gene expression levels were normalized to actin. The primer sequences used for PCR were: Nedd4 forward, 5′- CTTTATCCATTACCGACAG-3′ and Nedd4 reverse, 5′-GGTGGCTTCATCTTCTC-3′; VDAC2 forward, 5′-CTTCTTACAAGA GGGAGTG-3′ and VDAC2 reverse, 5′-GTCCCATCATTGACATTAG-3′; VDAC3 forward, 5′-TCTGGACCAACCATCTA-3′ and VDAC3 reverse, 5′-AGGCTGGCA TTATTTAC-3′; FOXM1 forward, 5′-GGAGGAAATGCCACACTTAGCG-3′ and FOXM1 reverse, 5′- TAGGACTTCTTGGGTCTTGGGGTG-3′; Actin forward,

5′-GCTCGTCGTCGACAACGGCT-3′ and Actin reverse, 5′-CAAACATGATCT GGCTCATCTTCTC-3′.

### Colony formation assay.
For the colony formation assay, cells were seeded in 60 mm dishes. At 70–80% confluent, cells were treated with DMSO or erastin for 24 h. Cells were trypsinized, counted, and re-plated in appropriate dilutions in six-well plate for colony formation. After 10–14 days of incubation, colonies were fixed and stained with a mixture of 6% glutaraldehyde (Amresco) and 0.5% crystal violet for 1 h. Remove the glutaraldehyde crystal violet mixture carefully and rinse with tap water. Leave the plates with colonies to dry in normal air at room temperature. Plating efficiency was determined for each cell line, and the surviving fraction was calculated based on the number of colonies that arise after treatment. Each experiment was repeated three times.

### Lentivirus and retrovirus production.
For lentivirus production, HEK293T cells were seeded in 100 mm dishes. At 70–80% confluent, cells were transfected with pCDH-CMV-MCS-EF1-Puro or pLKO.1-Puro vectors, together with pCMV-dR8.91 packaging plasmid and pCMV-VSV-G envelope plasmid at a ratio of 5:4:1 using the calcium phosphate transfection kit (Clontech). Twelve hours after transfection, the medium was replaced with fresh medium. Viral particles were collected 48 h after transfection, filtered with a 0.45 μm sterile filter, and concentrated by ultra-centrifugation at 4 °C (24000 rpm, 2 h, Beckman-Coulter ultracentrifuge XL-100K). Viral particles were resuspended in 2 ml fresh medium containing 8 μM per ml polybrene, and were plated with target cells in six-well plate. Then, the cell and viral particles mixture was centrifuged ($300 \times g$, 1 h) at room temperature to increase the infection efficiency. Cells were returned to the cell culture incubator for 24 h. Lentiviral-transduced cells were selected with medium containing 1 μg per ml pur-omycin for 7 days, and the medium was changed daily. For retrovirus production, HEK293-Amphotropic cells were transfected with pBabe-neo-Flag-Nedd4 and mutant plasmids using the calcium phosphate transfection kit. Viral particles were collected same as for the lentivirus. Cells were infected with retrovirus and selected with medium containing 500 μg per ml G418 for 7 days, and the medium was changed daily.

### In vitro ubiquitination assay.
The experiment was performed using in vitro ubiquitination assay Kit (Enzo, BML-UW9920-0001) according to the manu-facturer's instructions. Briefly, 5 mg of purified VDAC2 or VDAC3 proteins and 10 mg purified NEDD4 proteins were incubated with 0.5 mg E1 activating enzyme, 1.5 mg Ubiquitin, 0.5 mg various E2 enzymes UBCH4, UBCH5a, UBCH5b, UBCH5c, UBCH6 or UBCH7, and 2.5 mM ATP in reaction buffer (1.5 mM $MgCl_2$, 5 mM KCl, 1 mM dithiothreitol (DTT), 20 mM HEPES pH 7.4) in a total 20 ml reaction volume at 37 °C for 2 h. Ubiquitination on the substrate was then detected by WB analysis.

### Xenograph mouse model.
NU/NU Nude mice were purchased from Charles River (Beijing). All animal studies were performed following institutional guidelines of the Animal Care and Use Committee (IACUC) of Beijing Institute of Bio-technology. To generate murine subcutaneous tumors, melanoma cells ($5 \times 10^6$ cells per mouse) were injected subcutaneously into the left posterior flanks of 7-week-old immunodeficient female nude mice. Mice were monitored for the development of tumors by measurements of tumor weight, tumor length (L), and width (W); tumor volume based on calliper measurements was calculated by the modified ellipsoidal formula (tumor volume = $1/2(\text{length} \times \text{width}^2)$). When tumors reached a volume of approximately 50 $mm^3$, mice were randomly allocated into groups and treated with erastin via intraperitoneal injection for 20 or 30 days. Mice were then sacrificed; the tumor tissues were formalin-fixed and paraffin-embedded for histological analysis. The erastin was dissolved in 5% DMSO + corn oil (C8267, Sigma). To better dissolve erastin, we warm the tube at 37 °C water bath and shake it gently.

### Statistical analysis.
To ensure adequate power and decreased estimation error, we used large sample sizes and multiple independent repeats by independent investigators. In addition, multiple lines of experiments including different quantification methods were provided for consistent and mutually supportive results. The sample size was chosen according to well-established rules in the literature, as well as our ample previous research experience. Data are presented as mean ± SD. Unpaired student's $t$ tests were used to compare the means of two groups. One-way analysis of variance (ANOVA) was used for comparison among the different groups. When ANOVA was significant, post hoc testing of differences between groups was performed using the Least Significant Difference test. All data were analyzed by GraphPad Prism 6.0 (GraphPad Software, Inc.). A $p$ value < 0.05 was considered statistically significant.

### Reporting summary.
Further information on research design is available in the Nature Research Reporting Summary linked to this article.

## Data availability
The datasets obtained and analyzed during the current study are available from the corresponding authors upon reasonable request.

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

## Acknowledgements
We thank Dr. C.Y. Liang for providing reagents. This work was supported by the Technology Innovation Program of Beijing institute of technology and the National Natural Science Foundation of China (81772915) to Y.Y.

## Author contributions
Y.Y., M.L., and K.Z. performed the experiments and analyzed the data. T.G., F.Y., C.M., B.C., Y.S., and J.Z. participated in the data and sample collection. D.O. helped with the paper writing. J.Z. helped with paper discussion. W.C. and Y.Y. designed the experiments, analyzed the data, and wrote the paper.

## Competing Interests
The authors declare no competing interests.
