## [Peer Review File · Nature Communications]

Reviewers' comments:

Reviewer #1 (Remarks to the Author): Expert on melanoma and molecular biology

In this manuscript authors provide data to suggest that Nedd4 is the E3 ligase for VDAC2/3 limiting ferroptosis of melanoma cells following erastin stimuli. Overall, authors provide strong data to support main claims that (i) erastin induces ferroptosis (ii) that it is VDAC2/3-dependent and that (iii) Nedd4 is associated with VDAC2/3 causing its ubiquitination dependent degradation following erastin stimuli. Lastly, authors suggest that FOXM1 is the transcription factor responsible for activation of Nedd4 transcription and availability to limit VDAC2/3 availability. To substantiate authors conclusions the following points need to be addressed.

1. Initial data presented by authors suggest that VDAC2 and VDAC3 need to cooperate in order to induce ferroptosis following erastin stimuli. This is evident in Fig 1e, where overexpression of either alone has limited effect. Conversely, KD of either of the two was shown to be sufficient to affect degree of ferroptosis. This seemingly contradiction need to be addressed. Further and as important is the notion that most if not all subsequent experiments shown in this manuscript were performed with a single VDAC construct, and not with both, a point that needs to be further addressed experimentally.
2. Data suggesting that Nedd4 is the E3 ligase that limits VDAC2/3 stability (Fig 2/3) need to be substantiated by half-life studies for the endogenous proteins prior and following erastin addition, in the presence and absence of Nedd4.
3. Figure 4g-i demonstrate the effect of erastin on ferroptotic cell death in the presence of Nedd4 overexpression. Yet, the effect shown is rather modest and need to be discussed relative to the changes shown in GSH and Fe levels. Would scavengers for Fe and ROS ablate these effects?
4. Figure 5c-e claims for rescue of cell sensitivity to erastin upon OE of KR mutants of VDAC2/3 – yet the effect is really partial. The text in legend and results and related sections of the ms need to be properly adjusted.
5. Likewise, changes shown in Fig 6h-j are rather modest and claims should therefore be adjusted in respective sections in text.
6. Experiments performed in Fig 7a, 7c should be also carried out using constructs that will impair the levels of VDAC2/3 and FOXM1, in order to substantiate the model proposed by authors.
7. The authors claim for regulation of VDAC2/3 by erastin-FOXM1-Nedd4 in melanoma, while all studies are limited to two cell lines. How widely this phenomenon is truly seen in melanoma? are there subtypes of melanoma which are more amenable for this regulatory axis than others? BRAF or NRAS or PTEN mutant melanomas, for example?

Minor

Coding for numbers and colors shown in figures 3,4 appear randomly in the figure and should be noted also in legend.

Reviewer #2 (Remarks to the Author): Expert on ferroptosis

This manuscript describes a novel mechanism for the regulation of ferroptosis, an iron and lipid peroxide-dependent form of necrosis implicated in various diseases. The work was inspired by a

previously reported observation – VDAC2/3 expression decreases upon induction of ferroptosis by cystine transporter inhibitor, erastin (note that erastin has been reported previously to be able to bind to VDACS). Following this lead, by using several melanoma cell lines, the authors found that erastin, and possibly more generally, oxidative stress, upregulates transcription factor FOXM1; FOXM1 in turn upregulates the transcription of an E3 ubiquitin ligase Nedd4, which the authors demonstrated to be able to promote VDAC2/3 proteasomal degradation. The authors further showed that VDAC2/3 are positive regulators of ferroptosis, and thus FOXM1-Nedd4-VDAC2/3 form a negative feedback loop to regulate ferroptosis. Relevant to cancer, they showed Nedd4 inhibition can sensitize the anti-melanoma effect of erastin in both cell culture and xenograft experiments. Overall, the results are convincing and of high quality, the presentation is clear and logic, and the work is of general interest to related fields.

This reviewer has the following comments:

Major points:

1. Is the FOXM1-Nedd4-VDAC2/3 loop specific to erastin/cystine starvation or a general response to oxidative stress? This is an important question to address, since if the mechanism is validated to be a general one (testing a few additional stresses by western blot will do), its significance will certainly be much higher.
2. It is well known in the ferroptosis field that erastin is not effective in vivo due to issues including its rather poor solubility. This reviewer is curious how the authors could make it work (Fig. 7C) in this study when delivered peritoneally. The authors need to describe very clearly their experimental condition, such as how they reconstituted erastin for the injection, etc.
3. A thorough discussion of the potential link between the role of VDACS in mitochondria and their role in ferroptosis will be helpful, especially considering the recent publication in *Molecular Cell* "Role of mitochondria in ferroptosis". A convenient analysis monitoring the effect of VDAC2/3 knockdown on mitochondrial activity (change of mitochondrial membrane potential by using appropriate mitotracker; or change of oxygen consumption) will be important here.
4. Is the mechanism they describe here is due to the previous reported erastin-VDAC interaction? A careful discussion and clarification about this will be helpful and informative to the field.

Minor points:

1. Fig. 1a: ectopic VDACS did not show clear mitochondrial localization. Why is that?
2. Fig3b, 3d: shNedd4 without erastin treatment should be included as control; Fig. 3e: please double check the labeling.
3. Fig 4f, 4i: VDAC2/3 blot should be included
4. Fig 6c, d, e: detailed information about the reporter construction should be included, i.e., how do they get the sequence of FOXM BS1, BS2 (reported before or predicted by themselves)? How did they construct BS1/BS2 mutated reporter?
5. Fig 7: IHC of VDAC2/3, Nedd4, and a ferroptosis/redox marker (e.g., COX2) should be included for the tumor sections
6. For the in vivo xenograft model, the experimental procedure description is not consistent. In the Methods part, the author described that "To generate murine subcutaneous tumors, melanoma cells (5X10⁶ cells per mouse) were injected subcutaneously into the right posterior flanks of 7-week-old immunodeficient nude female mice", while in the figure legend, the author described that "C57BL/6 mice were injected subcutaneously with indicated A375 cells (2X10⁶ cells/mouse)". The inconsistency should be corrected.
7. In Discussion, line-279/280: ubiquitination of VDAC1 (?) by Nedd4?

Reviewer #3 (Remarks to the Author): Expert on melanoma

In this manuscript, the authors characterized the molecular mechanism by which ferroptosis activator erastin induces ubiquitination and proteasomal degradation of VDAC2/3 proteins in melanoma cells. The authors demonstrated that erastin induces the expression of Nedd4 E3 ligase, which directly interacts with and ubiquitinates VDAC2/3. In addition, the induction of Nedd4 by erastin was shown to be mediated by FOXM1-dependent and ROS-dependent transcriptional control of Nedd4 expression. This is an interesting study and may have therapeutic implications on

targeting ferroptosis in cancer cells. Overall, this study is well executed with sufficient experimental controls. Data are quite convincing and well presented. I only have a few comments, mostly moderate.

- 1) Suppression of ferroptosis by erastin through degradation of VDAC2/3 was proposed by the authors as a negative feedback mechanism to explain "erastin-induced cellular resistance". However, the melanoma cell lines used in this study appear to be quite sensitive to erastin (Fig. 7), and no significant treatment-induced resistance was presented. Additional melanoma cell lines with lower sensitivities to erastin could be used to test this hypothesis.
- 2) Does knockdown of Nedd4 affect the protein levels of VDAC2/3 in the absence of erastin (Figure 3b)?
- 3) The authors showed that Nedd4 regulated erastin-induced, but not RSL3-induced ferroptosis in melanoma cells (Fig. S4). Does RSL3 induce ROS-dependent activation of FOXM1 and Nedd4 expression in these cells?

Minor comments:

- 1) Fig S1d: '+' label for K48 in the last lane appears to be an error.
- 2) Fig 1c&d: combined interference with both VDAC2 and VDAC3 did not appear to induce a stronger effect as the authors stated (line 73), compared to interference with VDAC2 or 3 alone.
- 3) Errors in the legends for fig S4. RSL3 was supposed to be used in these experiments, instead of erastin.
- 4) A few other typos in the text, such as 'co-immunoprecipitation' (line 100) and 'protelysates' (line 131).

RESPONSE TO REVIEWERS

We are genuinely appreciative of the reviewer's constructive and insightful comments, according to which the manuscript has been carefully and rigorously revised. We hope the new version of our manuscript is now appropriately suited for publication in *Nature Communications*. A detailed response to the Reviewer's critiques and a description of the new experiments (in *italic*) follow:

Point-by-point Response to Reviewer #1

1. Initial data presented by authors suggest that VDAC2 and VDAC3 need to cooperate in order to induce ferroptosis following erastin stimuli. This is evident in Fig 1e, where overexpression of either alone has limited effect. Conversely, KD of either of the two was shown to be sufficient to affect degree of ferroptosis. This seemingly contradiction needs to be addressed. Further and as important is the notion that most if not all subsequent experiments shown in this manuscript were performed with a single VDAC construct, and not with both, a point that needs to be further addressed experimentally.

Response: We truly appreciate the reviewer's comments. Both VDAC2 and VDAC3 are expressed in the external membrane of the mitochondrion, and regulate the entry and exit of numerous ions and metabolites between the cytosol and the mitochondrion. The mitochondrion plays a crucial and proactive role in erastin-induced ferroptosis but not in GPX4 inhibition-induced ferroptosis¹. VDAC2 and VDAC3 may work together to mediate ferroptosis through regulating the homeostasis of different ions and metabolites in the mitochondrion. Knockdown of either VDAC2 or VDAC3 can disrupt the function of the mitochondrion, and suppress erastin-induced ferroptosis. However, overexpression of either VDAC2 or VDAC3 cannot promote erastin-induced ferroptosis, suggesting either VDAC2 or VDAC3 is necessary but not sufficient for keeping the sensitivity of melanoma cells to erastin. This phenomenon is also universal when two or more genes participate in a process together. Knockout of either one can disrupt the pathway and generate a mutant phenotype, while overexpression of either one is insufficient to activate the pathway, and two or more genes must be activated at the same time to produce the overexpression phenotype. Our results are also consistent with the previous studies². As shown in figure 3d,3e and supplementary figure S14,S15, knockdown of VDAC2 or VDAC3 in HT-1080 cells is sufficient to suppress the sensitivity of cells to erastin². Also, overexpression of VDAC3 in BJ-TERT cells yields no change in sensitivity to erastin².

We performed more function assay experiments with VDAC2 and VDAC3 constructs together. As shown in figure S8, we overexpressed VDAC2 and VDAC3 to rescue erastin-induced ferroptosis, which was suppressed by Nedd4 overexpression. Meanwhile, we also knocked down VDAC2 and VDAC3 simultaneously to suppress ferroptosis which enhanced by knockdown of Nedd4. Similarly, when we test whether VDAC2 and VDAC3 are essential for FOXM1 mediated ferroptosis, we simultaneously overexpressed or knocked down VDAC2 and VDAC3 (figure S10). In the main figures, there are also some function assay experiments with VDAC2 and VDAC3 constructs together. For example, when we use the wild type and mutants of VDAC2 and VDAC3 to rescue Nedd4 mediated ferroptosis, we transfected the cells with both VDAC2 and VDAC3 constructs (figure 5c-e, 5h). Similarly, we also knocked down both VDAC2 and VDAC3 to rescue FOXM1 mediated ferroptosis (figure 6h-j). However, because VDAC2 and VDAC3 are two different genes, we have to show their mRNA and protein expression levels separately, because we used different primers and antibodies. For example, when we detect the protein expression of VDAC2 and VDAC3 by confocal imaging, western blotting, and real-time PCR, we show the results separately. Similarly, in Co-IP and ubiquitination experiments, we have to detect the protein levels of VDAC2 and VDAC3 separately.

2. Data suggesting that Nedd4 is the E3 ligase that limits VDAC2/3 stability (Fig 2/3) need to be substantiated by half-life studies for the endogenous proteins prior and following erastin addition, in the presence and absence of Nedd4.

Response: We truly appreciate the reviewer's comments. We have completed the experiments suggested by the reviewer. The half-life of VDAC2/3 was reduced after erastin treatment, and knockdown of Nedd4 inhibited the degradation of VDAC2/3 either prior or following erastin treatment (figure S3c).

3. Figure 4g-I demonstrate the effect of erastin on ferroptotic cell death in the presence of Nedd4 overexpression. Yet, the effect shown is rather modest and need to be discussed relative to the changes shown in GSH and Fe levels. Would scavengers for Fe and ROS ablate these effects?

Response: We genuinely appreciate the reviewer's enthusiasm and great comments. In addition to the accumulation of lipid ROS, the intracellular Fe^{2+} also increased in ferroptosis^{3, 4}. Erastin-induced ferroptosis can be effectively inhibited by the iron chelators DFO, ciclopirox (CPX) and 2,2- bipyridyl (2,2-BP)^{3, 4}. As shown in figure S4a-d, DFO and CPX inhibited erastin-induced ferroptosis in Nedd4 overexpressing cells. GSH is essential for the lipid ROS clearance by glutathione peroxidase 4 (GPX4). The scavengers for lipid ROS, ferrostatin-1 (Fer-1) and Liproxstatin-1 (Lip-1), also can effectively inhibit erastin-induced ferroptosis in the present of Nedd4 overexpression (figure S4e-h). So scavengers for Fe^{2+} and lipid ROS can ablate the erastin-induced ferroptosis in the present of Nedd4 overexpression.

The reason for the modest effect of Nedd4 on erastin-induced ferroptosis in figure 4g-i may be that A375 and G-361 cells have activated RAF-MEK-ERK signaling and are sensitive to erastin. In mice xenograft experiments, the function of Nedd4 is more evident in MeWo cells (Wild type BRAF) than in A375 cells (figure 7 and figure S11). Because nearly half of melanoma patients carry BRAF mutants, we use A375 and G-361 cells in our experiments. Also, the effect of Nedd4 on ferroptosis is statistically significant in both A375 and MeWo cells. We have integrated these discussions into the revised manuscript.

4. Figure 5c-e claims for rescue of cell sensitivity to erastin upon OE of KR mutants of VDAC2/3 – yet the effect is really partial. The text in legend and results and related sections of the ms need to be properly adusted.

Response: Thank the reviewer for this comment. The rescue effect of KR mutants of VDAC2/3 is partial. One possible reason is that overexpressed Nedd4 may target other genes, which also regulate ferroptosis. As shown in figure 5c-e, overexpression of KR mutants of VDAC2/3 rescued more than half of the phenotypes mediated by Nedd4, suggesting VDAC2/3 is the most critical target of Nedd4 in erastin-induced ferroptosis. We genuinely appreciate the reviewer's excellent comments and have modified the related sections in the revised manuscript.

5. Likewise, changes in Fig 6h-j are rather modest and claims should therefore be adjusted in respective sections in text.

Response: Thank the reviewer for this comment. We have modified the related sections in the revised manuscript.

6. Experiments performed in Fig 7a, 7c should be also carried out using constructs that will impair the levels of VDAC2/3 and FOXM1, in order to substantiate the model proposed by authors.

Response: We truly appreciate the reviewer's enthusiasm and great comments. We have added the results in Figure S11a-d. Knockdown of FOXM1 or overexpression of VDAC2/3 has a similar function in erastin-induced ferroptosis and promotes the anti-tumor activity of erastin in mice xenograft model.

7. The authors claim for regulation of VDAC2/3 by erastin-FOXM1-Nedd4 in melanoma, while all studies are limited to two cell lines. How widely this phenomenon is truly seen in melanoma? are there subtypes of melanoma which are more amenable for this regulatory axis than others? BRAF or NRAS or PTEN mutant melanomas, for example?

Response: We genuinely appreciate the reviewer's enthusiasm and great comments. Although approximately 50% of clinical patients carry BRAF mutations, there are many other subtypes of melanoma cells which carry mutations on NRAS, PTEN, and TP53. To test whether Nedd4 and VDAC2/3 also regulate erastin-induced ferroptosis in other subtypes of melanoma, we performed xenograft experiments using MeWo, SK-MEL-2, WM2032, SK-MEL-3 and SK-MEL-24 cells. As shown in figure S11, S12 and S13, knockdown of Nedd4 or overexpression of VDAC2/3 has a similar function on tumor suppression in these melanoma cells. These results suggest that Nedd4 and VDAC2/3 play essential roles in regulating erastin-induced ferroptosis in different subtypes of melanoma cells.

Minor

1. Coding for numbers and colors shown in figures 3,4 appear randomly in the figure and should be noted also in legend.

Response: Thank the reviewer for this suggestion. Figure 4a-f share the same numbers and colors. Number 1 and black color represent control sh group; number 2 and red color represent Nedd4 sh + Vector group; number 3 and green color represent Nedd4 sh + Nedd4 WT group; number 4 and blue color represent Nedd4 sh + Nedd4 C867S group. Figure 4g-l share the same numbers and colors. Number 1 and black color represent vector group; number 2 and red color represent Nedd4 WT-Flag group; number 3 and green color represent Nedd4 C867S-Flag group. We have added the information into the revised manuscript.

Point-by-point Response to Reviewer #2

Specific concerns:

1. Is the FOXM1-Nedd4-VDAC2/3 loop specific to erastin/cystine starvation or a general response to oxidative stress? This is an important question to address, since if the mechanism is validated to be a general one (testing a few additional stresses by western blot will do), its significance will certainly be much higher.

Response: We truly appreciate the reviewer's enthusiasm and great comments. We treated the cells with RSL3 and H₂O₂, and found that both RSL3 and H₂O₂ increased the expression of FOXM1 and Nedd4. Similarly, the protein level of VDAC2/3 was also reduced after RSL3 or H₂O₂ treatment. Knockdown of FOXM1 or Nedd4 suppressed RSL3- and H₂O₂-induced protein degradation of VDAC2/3 (Figure S5a and figure below). However, VDAC2/3 does not affect RSL3-induced ferroptosis (figure S5b-e). The specific function of VDAC2/3 on erastin-induced ferroptosis may be caused by its role in mitochondria, which play a pivotal role in cysteine-deprivation induced ferroptosis but not in

*GPX4-inhibition induced ferroptosis*¹. Although other oxidative stress can activate the FOXM1-Nedd4-VDAC2/3 loop, it mainly regulates cysteine-deprivation induced ferroptosis.

2. It is well known in the ferroptosis field that erastin is not effective in vivo due to issues including its rather poor solubility. This reviewer is curious how the authors could make it work (Fig. 7C) in this study when delivered peritoneally. The authors need to describe very clearly their experimental condition, such as how they reconstituted erastin for the injection, etc.

Response: We appreciate this constructive comment made by the reviewer. The erastin is insoluble in water, but is soluble in DMSO. We purchased erastin from Selleckchem company (<http://www.selleckchem.com/products/erastin.html>). Selleck reports the solubility of erastin in DMSO is 19 mg/ml. Similarly, another company APExBIO shows that the solubility of erastin in DMSO is >27.4mg/mL (<https://www.apexbt.com/erastin.html>).

Erastin was used successfully in other mouse xenograft models in previous works^{5, 6}. For our in vivo experiments, we dissolved the erastin in 5% DMSO+Corn oil (Sigma C8267). The corn oil is used as a delivery vehicle for fat-soluble compounds^{7, 8, 9, 10}. To better dissolve it, we warmed the tube at 37 °C water bath and shook it gently. We have included this information in the methods.

3. A thorough discussion of the potential link between the role of VDACs in mitochondria and their role in ferroptosis will be helpful, especially considering the recent publication in Molecular Cell “Role of mitochondria in ferroptosis”. A convenient analysis monitoring the effect of VDAC2/3 knockdown on mitochondrial activity (change of mitochondrial membrane potential by using appropriate mitotracker; or change of oxygen consumption) will be important here.

Response: We thank the reviewer for this insightful comment. We have measured the mitochondrial membrane potential (MMP) using MitoTracker (tetramethylrhodamine ethyl ester, TMRE) in figure 5f-h. Knockdown of VDAC2/3 suppressed erastin-induced MMP hyperpolarization. In addition, overexpression of Nedd4 also suppressed erastin-induced MMP hyperpolarization, and VDAC2/3 mutants can rescue the inhibition effect of Nedd4. As mentioned in the Molecular Cell paper, MMP hyperpolarization increases lipid ROS generation, knockdown of VDAC2/3 suppresses MMP hyperpolarization and subsequent intracellular MDA level and ferroptotic cell death (figure 1c). There is a strong possibility of a potential link between the role of VDACs in mitochondria and their roles in ferroptosis, and we discussed it in the manuscript.

4. Is the mechanism they describe here is due to the previous reported erastin-VDAC interaction? A careful discussion and clarification about this will be helpful and informative to the field.

Response: We thank the reviewer for this great suggestion. It is a very interesting and important question whether erastin-induced VDAC2/3 degradation is due to the previous reported erastin-VDAC interaction. Unfortunately, there is no publication show which domain or amino acid mediates the interaction between erastin and VDAC, so we cannot figure out it in current work. If the key amino acids can be identified later, we can use the mutants to verify whether the erastin-VDAC interaction is

essential for VDAC degradation and the FOXM1-Nedd4-VDAC2/3 feedback mechanism. It is a great idea to discuss this possibility and definitely will be helpful to the ferroptosis field. We include this discussion in the revised manuscript.

Minor points:

1. Fig. 1a: ectopic VDACs did not show clear mitochondrial localization. Why is that?

Response: *Thank the reviewer for this comment. The expression pattern of VDACs in figure 1a is similar to the figure 4c of a previously published paper (see figure below)¹¹. Ectopic VDACs expressed in the cytoplasm and accumulated in some regions. The reason for this accumulation may be that the protein level of VDACs in ectopic experiment is higher than endogenous.*

2. Fig3b, 3d: shNedd4 without erastin treatment should be included as control; Fig. 3e: please double check the labeling.

Response: *We genuinely appreciate the reviewer's comments. We have answered the question and completed the experiments suggested by the reviewer. As shown in figure 3b and 3d, knockdown of Nedd4 without erastin treatment slightly increased the protein level of VDAC2/3 and reduced the ubiquitination of VDAC2/3. We have checked and corrected the labeling in figure 3e. The ubiquitination reaction only happened in the tube which contained all substrates and had 2 hours reaction time.*

3. Fig 4f, 4i: VDAC2/3 blot should be included

Response: *Thank the reviewer for this comment. Figure 4 shows the function of Nedd4 in ferroptosis. We suppressed the expression of Nedd4 using shRNA construct and transfected wild-type and mutant form of Nedd4. The figure 4f shows all the cell lines expressing expected levels of Nedd4. Then, we used these cell lines to test the function of Nedd4 in ferroptosis in figure 4b-e. The experiments in figure 4 mainly focus on the function of Nedd4 in ferroptosis, so we didn't include the expression data of VDAC2/3. We checked the protein level of VDAC2/3 and found that knockdown of Nedd4 slightly increased the expression of VDAC2/3 in cells without erastin treatment (see figure below). This data is consistent with the result in figure 3b. And overexpression of Nedd4 suppressed the protein level of VDAC2/3. The C867S mutant of Nedd4 did not affect the expression of VDAC2/3.*

4. Fig 6c, d, e: detailed information about the reporter construction should be included, i.e., how do they get the sequence of FOXM BS1, BS2 (reported before or predicted by themselves)? How did they construct BS1/BS2 mutated reporter?

Response: Thank the reviewer for this comment. We downloaded the promoter sequence of *Nedd4* from ensembl website (<http://www.ensembl.org/index.html>) and searched the FOXM1 binding site sequence TT(G/A)TT(G/C/A)(G/C). We found two FOXM1 binding sites in the promoter region of *Nedd4*. In addition, these two binding sites are also reported in a previous work¹². To generate the BS1/BS2 mutated constructs, we use the NEB Q5 site-directed mutagenesis kit (<https://www.neb.com/applications/cloning-and-synthetic-biology/site-directed-mutagenesis>), and use the online software to design primers (<http://nebasechanger.neb.com/>). We have added this information in the manuscript.

5. Fig 7: IHC of VDAC2/3, Nedd4, and a ferroptosis/redox marker (e.g., COX2) should be included for the tumor sections

Response: We appreciate this constructive comment made by the reviewer. We have finished the experiments suggested by the reviewer. As shown in Figure 7e, the IHC analysis of VDAC3, Nedd4, and 4-hydroxy-2-noneal (4HNE) of tumor xenograft samples was performed. The expression of Nedd4 is increased after erastin treatment. The protein level of VDAC3 is strongly suppressed by erastin in control sh cells, but not in Nedd4 sh cells. 4HNE is used as a marker for lipid peroxidation levels, which has been used in a recent work¹³. As shown in Figure 7e, the level of 4HNE is significantly increased after erastin treatment, knockdown of Nedd4 further enhanced the intracellular level of 4HNE. Because the VDAC2 antibody is not suitable for IHC, we only show the IHC staining of VDAC3.

6. For the in vivo xenograft model, the experimental procedure description is not consistent. In the Methods part, the author described that “To generate murine subcutaneous tumors, melanoma cells (5X10⁶ cells per mouse) were injected subcutaneously into the right posterior flanks of 7-week-old immunodeficient nude female mice”, while in the figure legend, the author described that “C57BL/6 mice were injected subcutaneously with indicated A375 cells (2X10⁶ cells/mouse)”. The inconsistency should be corrected.

Response: We apologize for the mistake and have corrected it.

7. In Discussion, line-279/280: ubiquitination of VDAC1 (?) by Nedd4?

Response: We truly appreciate the reviewer’s comments. VDAC1 cannot be ubiquitinated by

Nedd4. Our data only show that *VDAC1* also has the PPxY motif and binds to *Nedd4* (Figure S6a and S6b). The ubiquitination of R to K mutants of *VDAC1* mediated by *Nedd4* is increased when compared with the wild type *VDAC1* (Figure S7b). These results only suggest that *Nedd4* can ubiquitinate mutant forms of *VDAC1*. We have corrected the sentence in the discussion section.

Point-by-point Response to Reviewer #3

Specific concerns:

1) Suppression of ferroptosis by erastin through degradation of *VDAC2/3* was proposed by the authors as a negative feedback mechanism to explain “erastin-induced cellular resistance”. However, the melanoma cell lines used in this study appear to be quite sensitive to erastin (Fig. 7), and no significant treatment-induced resistance was presented. Additional melanoma cell lines with lower sensitivities to erastin could be used to test this hypothesis.

Response: We truly appreciate the reviewer’s comments. Tumor cells with activated RAS/RAF/MEK/ERK signaling are more sensitive to erastin treatment than non-activated cells². A375 cell carries homozygous *BRAF*(V600E) mutation, and G-361 cell is heterozygous. Approximately 50% of clinical patients carry *BRAF* mutations, so we use A375 and G-361 cells in our study. Both A375 and G-361 cells are more sensitive to erastin than MeWo cell which carries wild type *BRAF* gene (see figure below, unpublished data). Then, we tested whether knockdown of *Nedd4* could increase the anti-tumor activity of erastin in MeWo cells. As shown in figure S11e,f, because MeWo cell is less sensitive to erastin than A375 cell, the tumor volume of MeWo cell is larger than the tumor volume of A375 cell after erastin treatment. Knockdown of *Nedd4* or overexpression of *VDAC2/3* has a more obvious effect on tumor suppression in MeWo cells. Meanwhile, we also tested other melanoma cell lines SK-MEL-2, WM2032, SK-MEL-3, and SK-MEL-24. As shown in figure S12 and S13, knockdown of *Nedd4* or overexpression of *VDAC2/3* has similar effects on tumor suppression in these melanoma cells. And the effects on SK-MEL-2 and WM2032 cells are more evident than A375 cell. These results suggest that *Nedd4* and *VDAC2/3* regulate erastin-induced ferroptosis in different melanoma cells, and the tumor inhibition effect is most obvious in MeWo cells which carry wild-type *BRAF* genes.

2) Does knockdown of *Nedd4* affect the protein levels of *VDAC2/3* in the absence of erastin (Figure 3b)?

Response: Thank the reviewer for this comment. We have answered the question and completed the experiments suggested by the reviewer. As shown in Figure 3b, knockdown of *Nedd4* affects the protein levels of *VDAC2/3* in the absence of erastin, but the phenotype is stronger after erastin

treatment.

3) The authors showed that Nedd4 regulated erastin-induced, but not RSL3-induced ferroptosis in melanoma cells (Fig. S4). Does RSL3 induce ROS-dependent activation of FOXM1 and Nedd4 expression in these cells?

Response: We truly appreciate the reviewer's enthusiasm and great comments. RSL3 can induce the expression of FOXM1 and Nedd4, and suppress the protein level of VDAC2/3 in A375 cells (see figure below). However, VDAC2/3 is the membrane protein of mitochondria; it is sensitive to erastin-but not RSL3-induced ferroptosis. This result is consistent with a recent finding that mitochondria play a crucial role in cysteine-deprivation-induced ferroptosis but not in GPX4 inhibition-induced ferroptosis¹. We have discussed this result in the revised manuscript.

Minor comments:

1) Fig S1d: '+' label for K48 in the last lane appears to be an error.

Response: We apologize for the errors and have corrected it.

2) Fig 1c&d: combined interference with both VDAC2 and VDAC3 did not appear to induce a stronger effect as the authors stated (line 73), compared to interference with VDAC2 or 3 alone.

Response: We genuinely appreciate the reviewer's comments and have corrected it. Knockdown of VDAC2 and VDAC3 didn't show stronger effect on ferroptotic cell death phenotype, but the lipid ROS production and iron accumulation slightly reduced. So we changed the manuscript to "combined interference with VDAC2 and VDAC3 slightly reduced the lipid ROS production and iron accumulation". The possible reason for this result is that the cell viability of VDAC3-sh cell line is close to 100%, and knockdown both VDAC2 and VDAC3 could not show increased phenotype. To test our hypothesis, we increased the concentration of erastin from 5 μM to 10 μM, and found that combined interference with both VDAC2 and VDAC3 had a stronger effect on cell viability (see figure below).

3) Errors in the legends for fig S4. RSL3 was supposed to be used in these experiments, instead of erastin.

Response: We apologize for the errors and have corrected it.

4) A few other typos in the text, such as ‘co-immunoprecipitation’ (line 100) and ‘proteolysates’ (line 131).

Response: We apologize for the errors and have corrected it. We also carefully checked the writing in the revised manuscript.

Again, we thank the reviewers for his/her suggestions that have led to a much-improved manuscript.

Reference:

1. Gao M, Yi J, Zhu J, Minikes AM, Monian P, Thompson CB, *et al.* Role of Mitochondria in Ferroptosis. *Mol Cell* 2019, **73**(2): 354-363 e353.
2. Yagoda N, von Rechenberg M, Zaganjor E, Bauer AJ, Yang WS, Fridman DJ, *et al.* RAS-RAF-MEK-dependent oxidative cell death involving voltage-dependent anion channels. *Nature* 2007, **447**(7146): 864-868.
3. Dixon SJ, Lemberg KM, Lamprecht MR, Skouta R, Zaitsev EM, Gleason CE, *et al.* Ferroptosis: an iron-dependent form of nonapoptotic cell death. *Cell* 2012, **149**(5): 1060-1072.
4. Yang WS, SriRamaratnam R, Welsch ME, Shimada K, Skouta R, Viswanathan VS, *et al.* Regulation of ferroptotic cancer cell death by GPX4. *Cell* 2014, **156**(1-2): 317-331.
5. Xie Y, Zhu S, Song X, Sun X, Fan Y, Liu J, *et al.* The Tumor Suppressor p53 Limits Ferroptosis by Blocking DPP4 Activity. *Cell Rep* 2017, **20**(7): 1692-1704.
6. Zhu S, Zhang Q, Sun X, Zeh HJ, Lotze MT, Kang R, *et al.* HSPA5 Regulates Ferroptotic Cell Death in Cancer Cells. *Cancer Res* 2017.
7. Sangiorgi E, Capecchi MR. Bmi1 lineage tracing identifies a self-renewing pancreatic acinar cell subpopulation capable of maintaining pancreatic organ homeostasis. *Proc Natl Acad Sci U S A* 2009, **106**(17): 7101-7106.
8. Ertesvag A, Austenaa LM, Carlsen H, Blomhoff R, Blomhoff HK. Retinoic acid inhibits in vivo interleukin-2 gene expression and T-cell activation in mice. *Immunology* 2009, **126**(4): 514-522.
9. Alanis DM, Chang DR, Akiyama H, Krasnow MA, Chen J. Two nested developmental waves demarcate a compartment boundary in the mouse lung. *Nat Commun* 2014, **5**: 3923.
10. Xing YL, Roth PT, Stratton JA, Chuang BH, Danne J, Ellis SL, *et al.* Adult neural precursor cells from the subventricular zone contribute significantly to oligodendrocyte regeneration and remyelination. *J Neurosci* 2014, **34**(42): 14128-14146.
11. Gonzalez-Gronow M, Gomez CF, de Ridder GG, Ray R, Pizzo SV. Binding of tissue-type plasminogen activator to the glucose-regulated protein 78 (GRP78) modulates plasminogen activation and promotes human neuroblastoma cell proliferation in vitro. *J Biol Chem* 2014, **289**(36): 25166-25176.
12. Dai B, Pieper RO, Li D, Wei P, Liu M, Woo SY, *et al.* FoxM1B regulates NEDD4-1 expression, leading to cellular transformation and full malignant phenotype in immortalized human astrocytes. *Cancer Res* 2010, **70**(7): 2951-2961.
13. Zhang Y, Shi J, Liu X, Feng L, Gong Z, Koppula P, *et al.* BAP1 links metabolic regulation of ferroptosis to tumour suppression. *Nat Cell Biol* 2018, **20**(10): 1181-1192.

REVIEWERS' COMMENTS:

Reviewer #1 (Remarks to the Author):

authors have addressed reviewers comments and have thus notably improved the manuscript

Reviewer #2 (Remarks to the Author):

The authors have satisfactorily addressed all comments raised by this reviewer. A minor point they need to correct: although Ras-MEK signaling was originally proposed to be critical for ferroptosis, it was found later that Ras mutation has no impact to cancer cell sensitivity to ferroptosis (see Stockwell 2017 Cell paper), and the original observation is most likely due to the off-target effect of the used MEK inhibitor. Therefore, the discussion the authors added in the revision about Ras and ferroptosis should be deleted.

Reviewer #3 (Remarks to the Author):

The authors have addressed all my previous comments.